# Slow nucleosome dynamics set the transcriptional speed limit and induce RNA polymerase II traffic jams and bursts

**Robert C. Mines**[1], **Tomasz Lipniacki**[2]*, **Xiling Shen**[1,3,4]*

**1** Department of Biomedical Engineering, Duke University, Durham, North Carolina, United States of America, **2** Department of Biosystems and Soft Matter, Institute of Fundamental Technological Research, Polish Academy of Sciences, Warsaw, Poland, **3** Center for Genomic and Computational Biology, Duke University, Durham, North Carolina, United States of America, **4** Woo Center for Big Data and Precision Health, Duke University, Durham, North Carolina, United States of America

* tlipnia@ippt.pan.pl (TL); xiling.shen@duke.edu (XS)

**Data Availability Statement:** Code for the core algorithms and the calculation of simulation observables is available at SimTK (https://simtk.org/projects/histone_ddtasep) along with the

## Abstract

Nucleosomes are recognized as key regulators of transcription. However, the relationship between slow nucleosome unwrapping dynamics and bulk transcriptional properties has not been thoroughly explored. Here, an agent-based model that we call the dynamic defect Totally Asymmetric Simple Exclusion Process (ddTASEP) was constructed to investigate the effects of nucleosome-induced pausing on transcriptional dynamics. Pausing due to slow nucleosome dynamics induced RNAPII convoy formation, which would cooperatively prevent nucleosome rebinding leading to bursts of transcription. The mean first passage time (MFPT) and the variance of first passage time (VFPT) were analytically expressed in terms of the nucleosome rate constants, allowing for the direct quantification of the effects of nucleosome-induced pausing on pioneering polymerase dynamics. The mean first passage elongation rate $\gamma(h_c, h_o)$ is inversely proportional to the MFPT and can be considered to be a new axis of the ddTASEP phase diagram, orthogonal to the classical $\alpha\beta$-plane (where $\alpha$ and $\beta$ are the initiation and termination rates). Subsequently, we showed that, for $\beta = 1$, there is a novel jamming transition in the $\alpha\gamma$-plane that separates the ddTASEP dynamics into initiation-limited and nucleosome pausing-limited regions. We propose analytical estimates for the RNAPII density $\rho$, average elongation rate $v$, and transcription flux $J$ and verified them numerically. We demonstrate that the intra-burst RNAPII waiting times $t_{in}$ follow the time-headway distribution of a max flux TASEP and that the average inter-burst interval $\overline{t_{IBI}}$ correlates with the index of dispersion $D_e$. In the limit $\gamma \to 0$, the average burst size reaches a maximum set by the closing rate $h_c$. When $\alpha \ll 1$, the burst sizes are geometrically distributed, allowing large bursts even while the average burst size $\overline{N_B}$ is small. Last, preliminary results on the relative effects of static and dynamic defects are presented to show that dynamic defects can induce equal or greater pausing than static bottle necks.

associated data sets. Data sets are provided both in a .mat format that can be directly loaded into the MATLAB workspace. Alternatively, the files can be downloaded as Excel files in a .xlsx format and reprocessed by the reader.

**Funding:** XS and RCM were funded by the National Institutes of Health R35GM122465 (https://www.nih.gov/). RCM was also funded by the National Science Foundation Graduate Research Fellowship Program DGE-1644868 (https://www.nsfgrfp.org/). TL was funded by the Norwegian Financial Mechanism GRIEG-1 grant 2019/34/H/NZ6/00699 (operated by the National Science Centre Poland, https://ncn.gov.pl/eeanorwaygrants/calls/grieg?language=en). The funders had no role in study design, data collection and analysis, decision to publish, or preparation of the manuscript.

**Competing interests:** The authors have declared that no competing interests exist.

## Author summary

To perform specific functions, cells must express specific genes by copying the information in DNA into RNA via transcription. Structural proteins called nucleosomes are spaced every 200 base pairs along the length of a strand of DNA and play a crucial function in the regulation of gene activity by tightly binding DNA strands and condensing them into heterochromatin, preventing transcription by RNA polymerase II (RNAPII). Even on active genes where nucleosomes are loosely attached to DNA strands, the wrapping and unwrapping of nucleosomes pause transcription as RNAPII passes by. Previous mathematical models of transcription have compared this biological process to traffic on a one lane highway without obstructions. In contrast, our proposed model simulates transcription like traffic in a grid system where nucleosomes can be thought of as pedestrians or other vehicles crossing the road at regularly spaced intersections. Just as side street traffic and pedestrian crossings can cause cars to form convoys and cause jams limiting the max speed in an area, nucleosomes can cause RNAPII to form convoys that lead to bursts of mRNA production and limit the average polymerase flux through the gene.

## Introduction

Over the last 20 years, there has been a significant effort to explain stochasticity in molecular pathways focusing especially on the regulation of transcription and translation [1–6]. Most of the theoretical studies were concentrated on regulation of gene activity by means of binding and dissociation of transcription factors. However, a wide variety of assays have been developed to investigate epigenetic features that affect transcription by means of chromatin accessibility (ATAC-seq [7], DNASE-seq [8], FAIRE-seq [9]), histone modifications (Mint-ChIP [10] and ChIP-mentation [11]), or DNA methylation (bisulfite sequencing [12]). All of these epigenetic features exert their influence on transcription through the dynamics of nucleosomes, the histone octamers on which DNA is wrapped. Further, these epigenetic features exert measurable kinetic effects on the wrapping and unwrapping of DNA from the nucleosomes that may provide unexplored regulatory mechanisms for eukaryotic transcription [3,13].

At the molecular level, the process of transcription past a nucleosome is complex, but it has several key features: stalling of RNAPII on approach to a bound nucleosome, temporary nucleosome unwrapping/dissociation, and rebinding of the nucleosome if the site is not occupied by a polymerase. Histone modifications alter the duration and frequency of transcriptional pausing by influencing the fluctuation of nucleosomes between wrapped and unwrapped states (commonly referred to as "nucleosome breathing") [14,15]. The nucleosome's wrapping equilibrium and the absolute time scale of this process can potentially shape the transcriptional dynamics. Bintu et al. showed that transcription of bare DNA had an average of 4 pausing events per kbp with an average pause length of 4.4 seconds while transcription passing an unmodified nucleosome had an average of 14 pausing events per kbp with an average pause length of 10.2 seconds [16]. However, histone modification via acetylation significantly reduced the stalling to an average of 11 pausing events per kbp with an average pause length of 9.6 seconds [16]. In cases where the DNA is completely depleted of histones (as occurs with loss of stem-loop-binding protein), cells show a global increase in transcriptional elongation rates, loss of almost all transcriptional pausing as if the polymerase were transcribing bare DNA, and altered patterns of co-

transcriptional splicing [17]. Taken together, this data suggests that nucleosomes are the primary regulator of transcriptional elongation.

However, nucleosomes are also part of a large array of genetic and epigenetic features that control transcriptional elongation rates and the overall transcriptional flux [18,19]. In contrast to the dynamic nature of nucleosomes, there are also static defects associated with transcriptional pausing. The most critical static features that predict elongation slowdowns are high GC content, high exon density, and highly methylated CpG islands [18,19]. It is unintuitive whether these genetic and epigenetic features can best be described as static site defects or if they modify the dynamics of nucleosomes. Wang, Stein, and Ware demonstrated that nucleosomes have higher occupancy on regions of higher GC content and on exons [20]. Jonkers, Kwak, and Lis demonstrated that increased exon density increased RNAPII pausing due to co-transcriptional splicing at intron-exon junctions [18]. Collings, Waddel, and Anderson showed that methylated CpG sites (associated with transcriptional silencing) are more highly occupied by nucleosomes while unmethylated CpG islands are associated with polymerase recruitment and nucleosome depletion [21,22]. Therefore, these pausing events may be considered the result of either a static or dynamic defects.

While nucleosomes stall polymerases, polymerases also exert an effect on nucleosomes. At low rates, each polymerase must move past the nucleosome which rebinds between polymerase crossings. However, at high transcription rates, the lead polymerase pioneers its way through the nucleosomes, while holding the DNA open for its followers. This process keeps the nucleosome in the destabilized state longer, making it more likely to be totally displaced from the chromatin [23,24]. Thus, the nucleosomes induce convoy formation through pausing, but the resulting convoys obstruct nucleosome rebinding. In spite of the numerous simulations and extensive modeling of initiation-rate limited transcription and transcriptional bursting due to multi-state promoter dynamics, few computational and mathematical modeling studies have been performed to investigate the possibility of a transcriptional regime limited by nucleosome-induced pausing or the unique pattern of transcriptional bursting that this would induce [25–28]. Therefore, we developed a stochastic, agent-based model of transcription through nucleosomes arranged in the canonical beads on a string geometry. The nucleosomes undergo stochastic transitions from the wrapped to unwrapped state consistent with the thermodynamic equilibrium set by the microscopic rate constants. Encountering a nucleosome in the closed state causes polymerases to stall until the nucleosome reopens. In turn, the presence of polymerases prevents nucleosomes from closing.

From a theoretical perspective, the proposed model is an extension of a classical model in stochastic transport theory known as the Totally Asymmetric Simple Exclusion Process (TASEP) [29,30]. In the open boundary TASEP, particles can be injected through the first boundary (if the first lattice site is unoccupied) with propensity $\alpha$, removed through the other boundary (if the last lattice site is occupied) with propensity $\beta$, and advance with propensity $q$ (typically set to one to define the time scale) if there is no particle obstructing it on the next, non-boundary lattice site as shown in Fig 1A. Even though the TASEP only has nearest neighbor interactions, the TASEP exhibits a non-trivial kinetic phase diagram (Fig 1B) with the bulk density $\rho$, average hop rate $v$, and total particle flux $J = v \times \rho$ on the lattice completely defined by the initiation and termination propensities as shown in Eqs 1–3 [31–33].

$$v = \begin{cases} 1 - \alpha, & \alpha < \beta \text{ and } \alpha < {}^1/_2 \\ \beta, & \beta < \alpha \text{ and } \beta < {}^1/_2 \\ {}^1/_2, & \alpha, \beta \geq {}^1/_2 \end{cases} \tag{1}$$

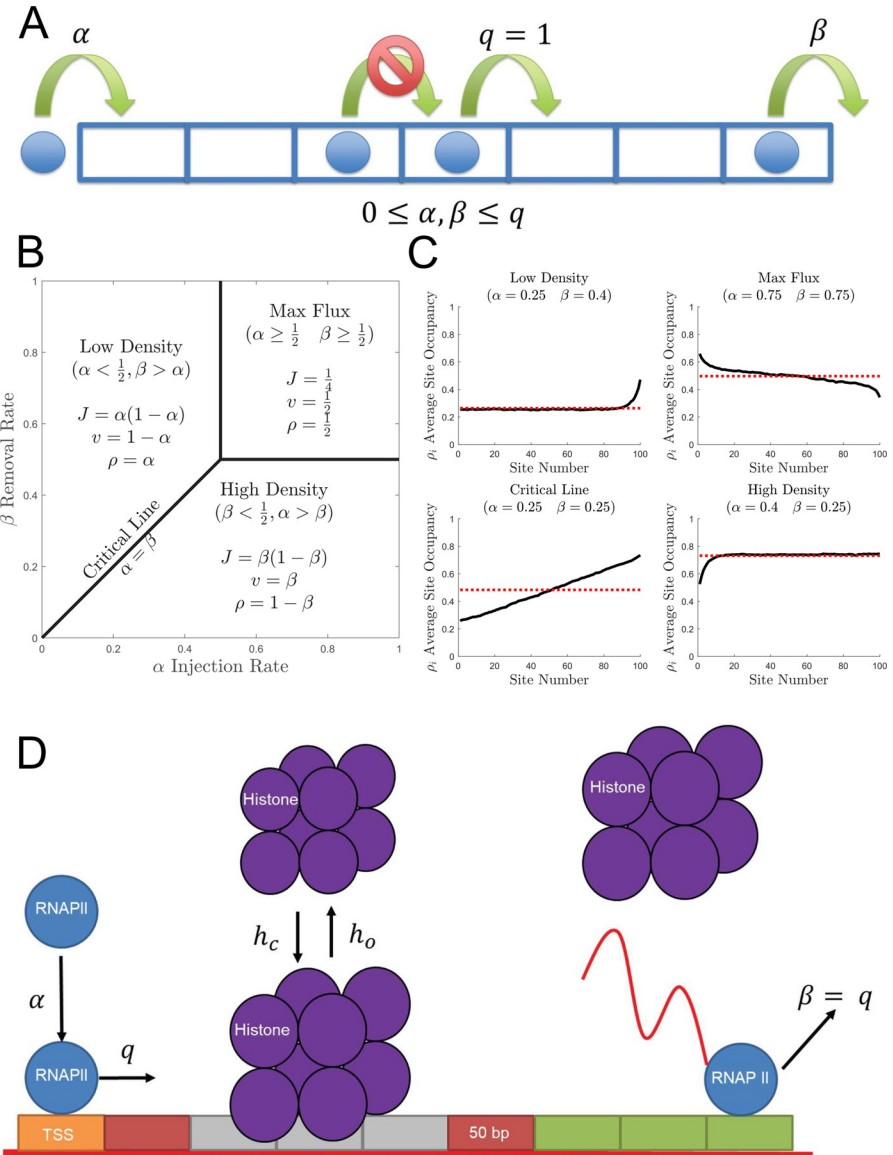

**Fig 1. Extension of the canonical TASEP to include nucleosomes as dynamic defects.** (A) Depiction of the canonical TASEP. Particles are injected into the first lattice site at rate α and advance with rate $q = 1$ when unobstructed by other particles. Particles are removed from the final lattice site at rate ß. (B) The canonical TASEP phase diagram in the αß-plane with the fundamental relationships for $J$, ρ, and $v$ summarized for each phase. (C) Simulated density profiles for each phase are shown by black lines, and dashed red lines indicate the mean-field theory prediction obtained for an infinitely long lattice. (D) Depiction of the proposed dynamic defect TASEP (ddTASEP) where periodically spaced nucleosomes function as extended-body dynamic defects. RNAPII bind the first lattice site (transcription start site, TSS) at rate α, RNAPII advances at rate $q$ when unobstructed by other polymerases or nucleosomes. RNAPII is then released from the gene at the end of transcription at rate ß, which is set equal to $q$ (ß = q). DNA in the nucleosome associated regions (containing 3 subsequent sites) unwraps from the nucleosome to the open conformation at rate $h_o$ and returns to the wrapped/closed state at rate $h_c$ when there are no polymerase present on the nucleosome associated sites. All lattice sites are 50 bp long.

$$\rho = \begin{cases} \alpha, & \alpha < \beta \text{ and } \alpha < {}^1\!/_2 \\ 1 - \beta, & \beta < \alpha \text{ and } \beta < {}^1\!/_2 \\ {}^1\!/_2, & \alpha, \beta \geq {}^1\!/_2 \end{cases} \qquad (2)$$

$$J = \begin{cases} \alpha(1 - \alpha), & \alpha < \beta \text{ and } \alpha < {}^1\!/_2 \\ \beta(1 - \beta), & \beta < \alpha \text{ and } \beta < {}^1\!/_2 \\ {}^1\!/_4, & \alpha, \beta \geq {}^1\!/_2 \end{cases} \qquad (3)$$

Additionally, the boundary defects give rise to a variety of density profiles as shown in Fig 1C.

The theory of the dynamic defect TASEP (ddTASEP), which is relevant to transcription, has not been fully developed. Van den Berg and Depken constructed a ddTASEP of transcription through single-site nucleosomes that would automatically be displaced from the gene after direct contact with polymerases [25]. Their model showed the first proof of concept of polymerase convoy organization and nucleosome depletion. However, their model treated polymerase advancement past nucleosomes as a single concerted step (i.e. RNAPII automatically moves through nucleosomes at a manually specified, slower rate rather than letting the wrapping dynamics dictate the advancement), the representative cases focused on low nucleosome density with slow rebinding, and they interpreted the results by mapping them on to a hypothetical two state model [25]. Waclaw et al. investigated a general dynamic defect TASEP with single site hopping particles and single site defects focusing primarily on closed loop cases with soft exclusion effects that allow nucleosome and polymerase co-occupancy with some coverage of the open boundary case [34].

Previous studies investigating the flux-density-hop rate ($J$-$\rho$-$v$) relationship of TASEPs with dynamic defects have focused primarily on weak or fast fluctuating perturbations and have utilized mean field approaches that cannot account for the long range particle-particle and particle-defect correlations [25,34,35–36]. Most notably, Waclaw et al. studied a variety of single site dynamic defect TASEPs with periodic and open boundaries with constrained and unconstrained defect binding [34]. However, most of their theoretical results were derived for the mathematically simpler case with periodic boundaries in which the bulk particle density remains constant.

In this study, we propose a transcription-dedicated, TASEP model with elongated, dynamic defects, and hard polymerase-nucleosome exclusion constraints (Fig 1D) that accounts for key features of RNAPII transcription through nucleosomes including nucleosome depletion, polymerase convoy organization, and transcriptional bursting. Additionally, we abandon the mean field approximation to focus on the regime in which transcription proceeds via propagation of polymerase convoys of arbitrary length. We derive the mean first passage time (MFPT) and variance of the first passage time (VFPT) of the pioneering polymerases that lead the convoys using a Markov Chain approach [37–39]. We quantified the strength of the nucleosome perturbation via the mean first passage rate $\gamma$, which is inversely proportional to the MFPT and depends on the microscopic nucleosome wrapping and unwrapping rate constants. The initiation rate $\alpha$ and the mean first passage rate $\gamma$ were then used to construct the ddTASEP phase diagram (orthogonal to the canonical TASEP $\alpha\beta$-phase diagram at $\beta = 1$) and to calculate the location of the nucleosome-induced jamming phase transition [29,31,32]. We identified two regimes in the novel $\alpha\gamma$-plane: initiation limited and nucleosome pausing (defect) limited. In both regions, we postulated approximate expressions for the elongation rate $v$, the bulk density $\rho$, and the bulk transcription flux $J$. Additionally, the inter-burst intervals and burst sizes generated by the slow nucleosome dynamics were investigated over a wide range of conditions spanning the initiation-limited and nucleosome-limited regimes. Finally, preliminary results on the effects of variability in the

RNAPII advance rate and in the nucleosome dynamic rate constants along DNA (e.g. caused by presence of exons or DNA methylation on CpG islands) are presented.

## Methods

### Numerical simulations

The model is simulated using a Gillespie algorithm with a variable number of reaction channels based on the set of all possible legal moves [40]. During each Monte Carlo step, the list of all legal polymerase and nucleosome moves is compiled. The propensities for each possible move are used to construct a categorical distribution, and a legal move is randomly selected from this distribution with probability proportional to its propensity. Next, the waiting time to the selected move is drawn from an exponential distribution with a mean waiting time given by the total propensity (sum of all legal propensities). The move is performed, and the system time is updated.

### Determination of ddTASEP properties from the simulation output

The canonical properties of the TASEP are the flux $J$, average hop rate $v$, and the average site occupancy/density $\rho$ [31,32]. The considered ddTASEP is ergodic and converges to a steady state probability density. For each Monte Carlo step, the state of the system is described by two vectors $\overrightarrow{s}$ and $\overrightarrow{h}$ composed of binary variables $s_i$ and $h_m$ indicating the polymerase occupancy of the $i^{th}$ site and the nucleosome occupancy of the $m^{th}$ nucleosome associated region, respectively. The averages of $v$, $\rho$, and $J$ are estimated within the time interval between the first passage time $t_{FP}$ (Monte Carlo step $N_{FP}$) and final time point $t_{MCS}$ (at least eight times greater than $t_{FP}$). All simulations were run for a fixed number of Monte Carlo steps ($N_{MCS} = 1.2 \times 10^6$ for qualitative trends or $N_{MCS} = 2 \times 10^6$ for quantitative results). The time averaged density profile $\rho_i$ is estimated in Eq 4.

$$\rho_i = \frac{1}{t_{MCS} - t_{FP}} \sum_{j=N_{FP}}^{N_{MCS}} s_i(t_j)\Delta t_j \tag{4}$$

As Derrida proved for the classic TASEP, far from the boundaries, the three primary properties of the TASEP converge exactly to values set by the microscopic rate constants for the initiation and termination rates ($\alpha$, $\beta$) [31,32]. For an infinitely long lattice, the boundary effects become negligible. For genes of 20 kbp (400 lattice sites) or longer, the boundary effects are weak, allowing the system to be characterized in terms of these three non-spatial, non-temporal metrics. Taking $N_{mRNA}$ to be the total number of mRNA produced, $N_{Sites}$ to be the number of lattice sites on the gene, and $\Delta t_{pass}^k$ to be the time between the $k^{th}$ RNAPII binding and terminating its transcription, the bulk averages of the three primary properties are defined in Eqs 5–7.

$$\rho = \frac{1}{N_{sites}} \sum_{i=1}^{N_{sites}} \rho_i \tag{5}$$

$$v = \frac{1}{N_{mRNA}} \sum_{k=1}^{N_{mRNA}} \frac{N_{sites}}{\Delta t_{pass}^k} \tag{6}$$

$$J = \frac{N_{mRNA}}{t_{MCS} - t_{FP}} \tag{7}$$

The bulk nucleosome densities $\rho_N$ and bulk nucleosome density profiles $\rho_{N,i}$ are calculated analogously to $\rho$ in Eq 1 and in Eq 2. However, the bulk nucleosome density $\rho_N$ is scaled by a factor of ¾ or more generally $\frac{k_N}{k_L + k_N}$ to account for the linker regions.

## Classifying burst events and measuring burst properties

When the pioneering polymerase reaches the end of the gene, the subsequent polymerases in the convoy exit with the flux predicted by the classical TASEP since a sufficiently long convoy functions as a shorter TASEP within the larger ddTASEP lattice. Given that the advance rate into the convoy and the termination rate are both greater than ½ (i.e. $q = 1$), polymerases exit with the highest flux predicted by the classical TASEP $J_{max} = ¼$. This implies that the average intra-burst waiting time $(\overline{t_{in}})$ is equal to 4 time units. However, there is no way to systematically predict what set of parameters $(\alpha, h_o, h_c)$ will give rise to bursting or what the optimal threshold is to classify an unknown waiting time $t_w$ as either an intra-burst waiting time $t_{in}$ or an inter-burst interval $t_{IBI}$.

Therefore, we take the following probabilistic approach. First, kernel density estimation is performed on the set of log-transformed waiting times $(\log_{10} t_w)$ weighted by $(\log_{10} t_w)$ so that the rare bursting events are more prominent. Next, an additional smoothing step is performed to reduce the noise from the probability density function estimation of the rare burst events. A local maxima search is performed to attempt to identify the intra-burst interval peak and the inter-burst interval peak. If only one peak is found, the ddTASEP is considered initiation limited, and no burst properties are recorded. If the system is bimodal, a threshold $T$ is calculated by taking the geometric mean of the peaks (which is also the arithmetic mean of the log transformed values). Burst events are defined by $t_w > T$. This threshold might filter out some intermediate cases, but it guarantees that those parameter sets $(\alpha, h_o, h_c)$ that generate robust bursts will be accurately classified while systematically eliminating false positives due to intrinsic stochasticity. The burst size $N_B$ is defined as the number of mRNA produced between these two breaks in transcription. Burst size and inter-burst intervals are averaged across simulations since they constitute rare events.

## Using correlations to quantify defect perturbation strength

The inter-site Pearson $r_{ij}$ correlation is defined below as

$$r_{ij} = \frac{\sum_{k=N_{sat}}^{N_{MCS}} [s_i(t_k) - \rho_i][s_j(t_k) - \rho_j]}{\sqrt{\sum_{k=N_{sat}}^{N_{MCS}} [s_i(t_k) - \rho_i]^2} \sqrt{\sum_{k=N_{sat}}^{N_{MCS}} [s_j(t_k) - \rho_j]^2}} \tag{8}$$

Eq 8 can be used to define a matrix $R = [r_{ij}]$ that contains the inter-site correlations for all site pairs $(i,j)$. Each row or column of the matrix $R$ is a vector of Pearson correlations that we denote as $\vec{r_i} = [r_{i1}, \cdots, r_{ii}, \cdots, r_{iN_{sites}}]$. Given the periodic nature of the lattice when a site is far from the boundaries, the inter-site correlations $r_{ij}$ should only be a function of the relative distance between the sites $(\Delta = j-i)$. Further, each central row vector $\vec{r_i}$ should exhibit the same qualitative features around the $i^{th}$ site as some hypothetical bulk correlation vector $\overline{r}$ (defined only in terms of inter-site distance) would around its central site at $\Delta = 0$. To construct the bulk correlation profile $\overline{r}$, we first chose a maximum distance $(\Delta_{max} = d)$ to evaluate the correlations over. Then, for each $i \in \{d, N_{sites} - d\}$, a new vector $r'_i = [r_{(i-d)i} \cdots r_{ii} \cdots r_{(i+d)i}]$ can be constructed that truncates each $\vec{r_i}$ and centers it around the $i^{th}$ site. Finally, we defined the bulk correlation profile $\overline{r}$ as the sliding window average over the vectors $r'_i$ as shown in Eq 9.

$$\overline{r} = [r_{-d} \cdots r_{-\Delta} \cdots r_0 \cdots r_\Delta \cdots r_d] = \frac{1}{2d+1} \sum_{i=d}^{N_{sites}-d} \vec{r'_i} = \frac{1}{2d+1} \sum_{i=d}^{N_{site}-d} [r_{(i-d)i} \cdots r_{ii} \cdots r_{(i+d)i}] \tag{9}$$

Each entry $r_\Delta$ of the vector $\overline{r}$ gives the average inter-site correlation based on relative distance $\Delta$. This metric allows for direct assessment of the strength and length scale of polymerase

cooperativity and also allows for more direct, visual comparison of the correlation strengths from different parameter sets than can be obtained from the correlation matrices themselves.

### Code implementation and code/data availability

Code was written in MATLAB R2018a (Math Works, Natick MA). High throughput parallel simulations were performed on a remote server (HARDAC at Duke Center for Genomic and Computational Biology) utilizing a SLURM job handler and an Lmod Environmental Module System that provided MATLAB R2017b. Given that MATLAB's internal parallelization could not be used due to conflict between it and the job scheduler, each instance of the parameter sweep was evaluated on a separate node initialized into the same environment. Since the environmental handler initialized with the same default random number seed for each iteration, random number seeds were specified manually.

Code for the core algorithms and the calculation of simulation observables is available at SimTK (**https://simtk.org/projects/histone_ddtasep**) along with the associated data sets. Data sets are provided both in a.mat format that can be directly loaded into the MATLAB workspace. Alternatively, the files can be downloaded as Excel files in a.xlsx format and reprocessed by the reader.

## Results

### ddTASEP transcription model formulation

In the proposed model (Fig 1D), the ddTASEP lattice is segmented into 50 base pair (bp) sites. A single nucleosome occupies three sites, and there is one linker site between each nucleosome-associated region. This discretization was chosen since the RNAPII footprint on DNA is generally estimated to be roughly 40 bp [41,42]. Further, the nucleosome linkers range in size from 10–70 bp [43]. Additionally, DNA wrapped around nucleosomes is known to be exactly 145–147 bp [43]. Last, assays such as BruDRB-seq have suggested that the maximum velocity of polymerases is roughly 50 bp/s even for extreme outliers across multiple cells types [19], and other more common assays such as MS2-tagging have suggested transcription speeds of 26–86 bp/s [44]. Thus, 50 bp sites will be occupied by one RNAPII enzyme which will take on average one second to clear the site on bare DNA at 50 bp/s.

There are three possible actions for polymerases: initiation, advancement, and termination. If the transcription start site (TSS) is open, polymerases can be initiated with propensity $\alpha$. Once on the gene, RNAPII advances with propensity $q$ while it is not obstructed by another RNAPII or a nucleosome in the wrapped/closed state. It must pause until the nucleosome or other polymerase moves out of its way. Nucleosomes transition from the open (unwrapped) to the closed (wrapped) position with propensity $h_c$ when all three nucleosome associated sites are empty. The nucleosome unwraps to the open position with propensity $h_o$. At the end of the gene, the RNAPII will terminate with propensity $ß = q$.

Since the lattice is segmented into 50 bp sites and the characteristic RNAPII velocity is of order of 50 bp/s, the RNAPII advance rate (from one 50 bp site to the next) $q$ is on the order of 1 site/s. For the sake of simplicity, we will use time units such that $q = 1$ in the figures. However, $q$ is left in the equations associated with the figures for generality. In the last section of this manuscript, the role of non-uniformity of DNA (due to the presence of exons, methylated CpG islands, and high GC content) will be incorporated by assigning each lattice site a new nominal advance rate $q_i$ or by assigning each nucleosome a unique set of rate constants $h_{c,m}$ and $h_{o,m}$.

The model can be described formally as follows. Let $s_i$ be a binary variable representing the polymerase occupancy of the $i^{th}$ lattice site and let $h_m$ be the nucleosome occupancy of the

nucleosome associated with sites $i$, $(i+1)$, $(i+2)$, *and* $(i+3)$. Site $i$ is a linker site, and the others are the nucleosome associated sites. When any of $(i+1)$, $(i+2)$, *and* $(i+3)$ are occupied, nucleosome $m$ cannot rebind. These rules are summarized as follows:

$$\text{Initiation}: \ s_1 = 0 \xrightarrow{\alpha} s_1 = 1 \tag{10}$$

$$\text{Nucleosome Entry}: \ (s_i = 1, s_{i+1} = 0, h_m = 0) \xrightarrow{q_i} (s_i = 0, s_{i+1} = 1, h_m = 0) \tag{11}$$

$$\text{Normal Advancement}: \ (s_{i+1} = 1, s_{i+2} = 0) \xrightarrow{q_{i+1}} (s_{i+1} = 0, s_{i+2} = 1) \tag{12}$$

$$\text{Termination}: \ s_N = 1 \xrightarrow{\beta} s_N = 0 \tag{13}$$

$$\text{Wrapping}: \ (s_{i+1} = 0, s_{i+2} = 0, s_{i+3} = 0, h_m = 0) \xrightarrow{h_{c,m}} (s_{i+1} = 0, s_{i+2} = 0, s_{i+3} = 0, h_m = 1) \tag{14}$$

$$\text{Unwrapping}: \ h_m = 1 \xrightarrow{h_{o,m}} h_m = 0 \tag{15}$$

## Nucleosome dynamics control polymerase convoy formation and transcriptional bursting

The dynamical defect TASEP exhibits a wider array of dynamical behavior than the classical TASEP [31,32] or static defect TASEP [45–48]. The pausing induced by nucleosome binding leads to organization of RNAPII into convoys. In Fig 2, we analyze the behavior of the considered ddTASEP.

The mRNA accumulation curves shown in Fig 2A all exhibit characteristic features. First, there is a delay due to the first passage time for the first RNAPII to clear the length of the gene as it interacts with the nucleosomes. Second, there is an approximately linear increase in the number of transcriptional events after this initial period. In the weakly perturbed cases (bottom row), the polymerases advance quasi-deterministically giving rise to uninterrupted, linear in time mRNA synthesis. As the nucleosome dynamics slow down (top row), two patterns of nucleosome-induced pausing are observed. In the slow opening and closing case ($h_o = h_c = 0.001$, top left), periods of sustained transcription are disrupted by nucleosome binding resulting in large transcriptional pauses. Therefore, the nearly vertical sections of mRNA production in these cases represent mRNA bursts observed when a convoy of RNAPII reaches end of the gene. In the fast closing and slow opening cases ($h_o = 0.001$, $h_c = 0.1$, top right), fast nucleosome re-wrapping leads to the formation of smaller, tight burst groups without the long, uninterrupted periods of continuous transcription seen in the previous case.

The self-organization of polymerases into convoys that give rise to the bursts in Fig 2A is also readily apparent in the kymographs in Fig 2B. In the less nucleosome perturbed cases (bottom row), the kymographs show the polymerases moving quasi-deterministically along the length of the gene with few collisions and limited convoy formation as would be expected in a canonical TASEP. As the nucleosome-induced pausing becomes stronger in the upper subplots of Fig 2B, the polymerases organize into well separated convoys, most notably in the fast binding and slow opening case ($h_o = 0.001$, $h_c = 0.1$, top right). Finally, in the extremely slow opening and closing case ($h_o = h_c = 0.001$, top left), both the completely empty gene and completely polymerase-occupied configurations of the lattice are observed.

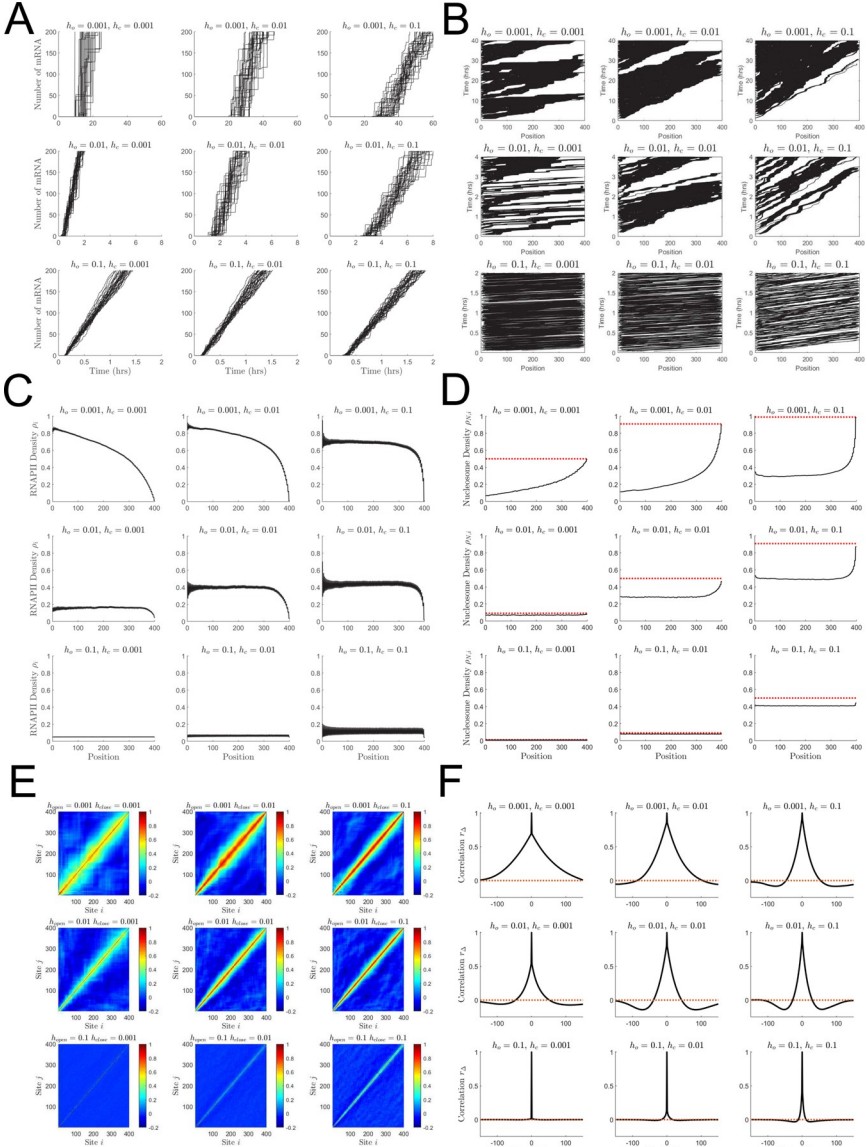

**Fig 2. Qualitative behavior of the ddTASEP over a range of representative parameters.** All sub-figures are composed of 9 subplots. For all plots, the transcription initiation rate $\alpha$ was set to 0.05, while $h_o$ is held constant along rows (increasing from 0.001→0.01→0.1 from top to bottom) and while $h_c$ is held constant along columns (increasing from 0.001→0.01→0.1 from left to right). The top right subplot deviates the furthest from the canonical TASEP behavior while the bottom left most closely resembles ideal TASEP behavior. Simulations shown in panels A, C, D and F utilized $4.5 \times 10^5$ Monte Carlo Steps with 50 replicates. (A) Number of completed mRNAs as a function of time. Notice that the simulation times for the top, middle, and bottom rows are 60, 8, and 2 hours, respectively. (B) Kymographs plotting the position of each RNAPII on the gene (x-axis) as time advances (y-axis). (C) Time averaged RNAPII site density $\rho_i$ as a function of position. The wavy density profiles are caused by the fact that the linker sites have a higher average RNAPII occupancy than the neighboring nucleosome sites. (D) Time averaged nucleosome site density $\rho_{N,i}$ (black line) as a function of position (ignoring linker sites). The bulk nucleosome density is anti-correlated with the polymerase density, reaching a maximum at the end points where polymerases are rapidly ejected. The dotted red lines indicate the resting-state nucleosome density prior to transcriptional initiation. (E) Inter-site correlation heatmaps with colors bars ranging from -0.2 to 1. (F) Average inter-site correlation profile ($r_\Delta$, black line) as a function of relative inter-site distance $\Delta$ with the dotted red line indicating zero.

Based on Eq 2, a canonical TASEP with $\alpha = 0.05$ would be in the initiation limited region and accordingly have $\rho = 0.05$. However, while all the ddTASEP density profiles shown in Fig 2C resemble the initiation limited region from Fig 1C, the values for the bulk density $\rho$ are always equal or larger than those of the classical TASEP. Further, the density profiles $\rho_i$ showed a sawtooth wave pattern along the spatial axis since the linker sites always had higher polymerase occupancy due to nucleosome induced pausing. As expected, higher levels of polymerase occupancy lead to noticeable nucleosome depletion as demonstrated by the difference between resting probabilities of nucleosome occupancy (dotted red line) and the steady state nucleosome density profiles in Fig 2D. The nucleosome occupancy profiles presented here look like inverted versions of the polymerase density profiles consistent with the hard exclusion constraint that allows polymerases to hold nucleosomes in the open conformation.

Last, the central sites in the inter-site correlation heatmap Fig 2E confirm our expectation that the inter-site correlations only depend on the relative distance between sites. Using Eq 9, for inter-site correlation $r_\Delta$ in Fig 2F we directly quantify the strength of cooperativity between polymerases at distance $\Delta$. In the highly nucleosome perturbed case (with slow opening and closing, $h_o = h_c = 0.001$, top left), the positive correlation spans over a distance comparable with the gene length suggesting that polymerase convoys for a given set of $h_c$ and $h_o$ could be longer than the length of the gene. The intermediate cases (most notably the central case where $h_o = h_c = 0.01$) exhibit both a short range positive correlation associated with RNAPII convoys and a longer range negative correlation showing that convoys are separated by regions of rebinding nucleosomes. In the case of the weak nucleosome binding ($h_o = 0.1$, $h_c = 0.001$ and $h_o = h_c = 0.1$), no inter-site correlation is observed showing that bursts are not formed.

## Mean and variance of first passage time

As demonstrated in Fig 2A and 2B, our model implies that the time needed to produce first complete mRNA depends on the nucleosome-induced pausing. Using Markov chain theory, the expected passage time through a nucleosome unit (consisting of one linker site and three nucleosome sites) for the first polymerase can be calculated [49,50]. Due to the periodic spacing of nucleosome binding sites, the MFPT to produce an mRNA after gene reactivation is simply the sum of the expected times for the first polymerase to clear each nucleosome unit.

The interaction between a polymerase on a linker site and a nucleosome can be treated as a three state Markov Chain (Fig 3A), where State 1 (polymerase on linker with nucleosome open) and State 2 (polymerase on linker with nucleosome closed) can reversibly communicate until the absorbing State 3 (polymerase enters the first nucleosome site) is reached as shown in Fig 3A. The evolution of the state probability distribution of this Markov Chain $\overrightarrow{p} = [p_1 \quad p_2 \quad p_3]$ can be described as a system of Forward Kolmogorov Equations with generator matrix $Q$ [37,39,49]:

$$\frac{d\overrightarrow{p}}{dt} = \overrightarrow{p}Q = [p_1 \quad p_2 \quad p_3] \begin{bmatrix} -(q + h_c) & h_c & q \\ h_o & -h_o & 0 \\ 0 & 0 & 0 \end{bmatrix} \tag{16}$$

All of the moments of the first passage time of any absorbing Markov chain are uniquely determined by the submatrix of transient states in Q which will be referred to as $R$.

$$R = \begin{bmatrix} -(q + h_c) & h_c \\ h_o & -h_o \end{bmatrix} \tag{17}$$

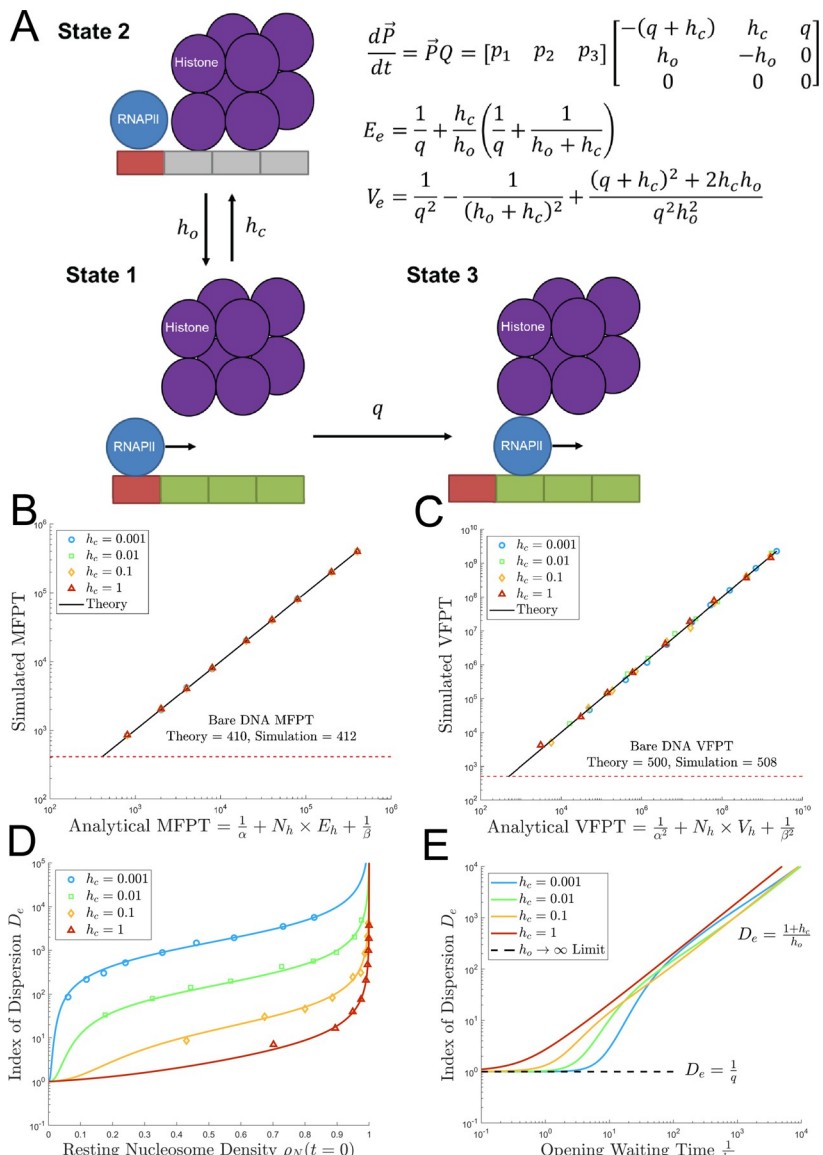

**Fig 3. Deriving and validating the mean and variance of first passage time equations.** (A) A state transition diagram utilized to calculate the mean ($E_e$) and variance ($V_e$) of first passage time to enter a nucleosome. Each arrow is labeled with the rate constant for the transition between the neighboring states. (B) Log-log plot of simulated mean first passage time (Eq 22) against the predicted analytical expression for mean first passage time. The black line represents the theoretical prediction. The dashed red line represents the Bare DNA limit obtained by simulations with $h_c = 0$. (C) Log-log plot of simulated variance of first passage time against the predicted analytical expression for VFPT (Eq 26) with the Bare DNA limit represented by the dashed red line. (D) Plot of index of dispersion of the waiting time to enter a nucleosome $D_e = V_e/E_e$ against the resting state binding probability of the nucleosomes ($\rho_N = \frac{h_c}{h_o+h_c}$). $h_c$ is set to be 0.001 (blue), 0.01 (green), 0.1 (orange), and 1 (red). $h_o$ was adjusted to achieve $\gamma \in \{0.002, 0.005, 0.01, 0.02, 0.05, 0.1, 0.2, 0.5, 0.99\}$. Solid lines indicate the theoretical prediction given by Eq 27. (E) Log-log plot of the analytical expression for the index of dispersion of the waiting time to enter a nucleosome $D_e$ (Eq 27, analytical) as a function of the expected waiting time for nucleosome opening ($1/h_o$). The dashed black line shows the index of dispersion in the limit of $h_o \to 0$.

The fundamental Matrix $N$ is then given by [38,49]

$$N = -R^{-1} = \begin{bmatrix} \dfrac{1}{q} & \dfrac{1}{q}\left(\dfrac{h_c}{h_o}\right) \\[3mm] \dfrac{1}{q} & \dfrac{1}{h_o} + \dfrac{1}{q}\left(\dfrac{h_c}{h_o}\right) \end{bmatrix} \tag{18}$$

The expected waiting times for the polymerase to enter the nucleosome site from the open state ($E_o$, State 1) and closed state ($E_c$, State 2) can be expressed in terms of the fundamental matrix $N$ as

$$\begin{bmatrix} E_o \\ E_c \end{bmatrix} = N\vec{1} = \begin{bmatrix} \dfrac{1}{q} & \dfrac{1}{q}\left(\dfrac{h_c}{h_o}\right) \\[3mm] \dfrac{1}{q} & \dfrac{1}{h_o} + \dfrac{1}{q}\left(\dfrac{h_c}{h_o}\right) \end{bmatrix}\begin{bmatrix} 1 \\ 1 \end{bmatrix} = \begin{bmatrix} \dfrac{1}{q}\left(1 + \dfrac{h_c}{h_o}\right) \\[3mm] \dfrac{1}{h_o} + \dfrac{1}{q}\left(1 + \dfrac{h_c}{h_o}\right) \end{bmatrix} \tag{19}$$

The overall expected waiting time to enter the nucleosome associated site ($E_e$) is given by the weighted average of $E_o$ and $E_c$ with respect to the probability of finding the nucleosome in the open or closed state upon the polymerase's initial approach $\left(\vec{p_0} = \begin{bmatrix} \dfrac{h_o}{h_o + h_c} & \dfrac{h_c}{h_o + h_c} \end{bmatrix}\right)$.

$$E_e = \vec{p_0}N\vec{1} = \begin{bmatrix} \dfrac{h_o}{h_o + h_c} & \dfrac{h_c}{h_o + h_c} \end{bmatrix}\begin{bmatrix} E_o \\ E_c \end{bmatrix} = \dfrac{1}{q} + \dfrac{h_c}{h_o}\left(\dfrac{1}{q} + \dfrac{1}{h_c + h_o}\right) \tag{20}$$

The expected passage time to transcribe past a nucleosome and its three associated sites ($E_h$) is the sum of $E_e$ and the expected time to pass through 3 empty sites ($3/q$)

$$E_h = \dfrac{3}{q} + E_e = \dfrac{4}{q} + \dfrac{h_c}{h_o}\left(\dfrac{1}{q} + \dfrac{1}{h_c + h_o}\right) \tag{21}$$

Consequently, the overall MFPT to transcribe the gene is given by

$$MFPT = \dfrac{1}{\alpha} + N_h E_h + \dfrac{1}{\beta} \tag{22}$$

where $\alpha$, $\beta$ and $N_h$ are the initiation rate, termination rate, and the number of nucleosome units, respectively. Fig 3B shows that MFPT formula (Eq 22) accurately predicts the simulated MFPTs ($R^2 = 0.99995$), generated by varying $h_o$ to attain various values of $\gamma$ across a log scale from 0.001 to 1 while holding $h_c$ constant.

We use $E_h$ also to construct a new parameter $\gamma$ called the mean first passage elongation rate.

$$\gamma = \dfrac{4}{E_h} \tag{23}$$

By definition $\gamma = 1$ when the nucleosomes are constantly open $h_c = 0$. The reader should note the distinction between the bulk elongation rate $v$ (the average velocity of all polymerases to transcribe a gene) and the mean first passage rate $\gamma$ (the expected/ensemble-averaged velocity of the first polymerase to transcribe a gene).

Another useful metric to understand the effects of the nucleosomes on transcription is the variance of the first passage time (VFPT). The variance of the nucleosome entrance waiting

time ($V_e$) was calculated with the following equation from [38].

$$V_e = 2\overrightarrow{p_o}(N^2)\overrightarrow{1} - E_e^2 = \frac{1}{q^2} - \frac{1}{(h_o + h_c)^2} + \frac{(q+h_c)^2 + 2h_ch_o}{q^2h_o^2} \qquad (24)$$

As with the mean first passage time, the variance of the time to pass through a nucleosome unit ($V_h$) and the overall variance of the first passage time (VFPT) are obtained from $V_e$.

$$V_h = \frac{3}{q^2} + V_e = \frac{4}{q^2} - \frac{1}{(h_o + h_c)^2} + \frac{(q+h_c)^2 + 2h_ch_o}{q^2h_o^2} \qquad (25)$$

$$VFPT = \frac{1}{\alpha^2} + N_hV_h + \frac{1}{\beta^2} \qquad (26)$$

Fig 3C demonstrates that the analytical expression for VFPT (Eq 26) accurately predicts the simulated VFPTs ($R^2 = 0.991$), over three orders of magnitude of $h_o$ and $h_c$. When nucleosome binding was turned completely off ($h_c = 0$), the analytical expression and simulation both provided the expected result for bare, nucleosome-free DNA (i.e. a canonical TASEP).

Finally, we introduce the index of dispersion $D_e = \frac{V_e}{E_e}$, that can be used to measure the deviation of the ddTASEP's behavior from a classical TASEP on nucleosome-free DNA for which $D_e = \frac{1}{q}$. The index of dispersion of the time to enter a single nucleosome $D_e$ was found to be

$$D_e = \frac{V_e}{E_e} = \frac{1}{q} + \frac{q+h_c}{qh_o} + \frac{1}{h_o+h_c} - \frac{2(h_o+h_c)}{h_o^2 + h_c^2 + h_c(q+2h_o)} \approx \begin{cases} \frac{1}{q}, & h_c = 0 \ or \ h_o \to \infty \\ \frac{q+h_c}{qh_o}, & h_o \to 0 \end{cases} \qquad (27)$$

In the limits as $h_c = 0$ or $h_o \to \infty$, the index of dispersion converges to $(1/q)$ which is consistent with the canonical TASEP limit since this is simply the index of dispersion for an exponential distribution or a Poisson process [49]. In contrast, when $h_o \ll h_c$, the index of dispersion becomes dominated by the expected waiting time for the nucleosome to reopen.

Fig 3D and 3E investigate whether the index of dispersion is more strongly controlled by the resting nucleosome density (Fig 3D) or the absolute time scale of the reopening time $\frac{1}{h_o}$ (Fig 3E). As shown in Fig 3D, there is a linear increase in index of dispersion with respect to increasing resting nucleosome density ($\frac{h_c}{h_o+h_c}$) followed by an abrupt increase as the resting density goes to one. It is also possible that this change in the index of dispersion likely arises from the associated decrease in $h_o$ needed to achieve the desired initial nucleosome density for fixed $h_c$. Fig 3E shows a smooth transition from the quasi-deterministic random walk of the classical TASEP ($D_e \approx 1/q$ in the limit $h_c = 0$) to the nucleosome limited cases where the unwrapping waiting time associated with nucleosome breathing dominates the index of dispersion $D_e \approx (1+h_c)/h_o$ in the limit $h_o \to 0$. Thus, these large deviations from canonical TASEP behavior leading to polymerase self-organization and bursting are driven primarily by infrequent interactions with nucleosomes with slow re-opening (low $h_o$) but can be marginally enhanced by more frequent interactions with nucleosomes (higher resting nucleosome density) due to higher rates of re-wrapping (high $h_c$).

## Validity of Coarse-Graining the Lattice

A core assumption of this model is that the lattice can be treated in terms of 50 bp sites instead of at 1 bp resolution (consistent with the fact that RNAPII advance one nucleotide at a time). It is a well-known result that, for fixed lattice size, a TASEP with particles of length $L > 1$ will

have reduced flux through the lattice relative to a TASEP with particles of length $L = 1$ [51,52]. However, the effect of simultaneously changing both particle length and lattice size on transcription flux is not intuitive.

Therefore, we considered the possibility of using non-exponential waiting times. The true waiting time distribution for this system is likely an $Erlang(k = n, \lambda = n)$ distribution with $n \in [1, 50]$. Two special cases of this exist: When $n = 1$, the canonical exponential waiting time distribution is reobtained. When $n = 50$, the interpretation is that all 50 nucleotide polymerization steps occur with waiting times given by independent and identically distributed exponential variables with a uniform rate of 50 bp/s. (Both cases represent extremes that are not likely to be biologically realistic.) The variance of the advancement waiting time under the exponential distribution is $1s^2$ while it is $0.02s^2$ under the $Erlang(k = 50, \lambda = 50)$ distribution. In a non-Markovian TASEP without defects, the reduced variance in the hop time leads to fewer collisions allowing the average (bulk) particle hop rate $v$ to approach the nominal hop rate $q$. This leads to substantially enhanced flux $J$ relative to a Markovian TASEP with exponentially distributed waiting times [53].

However, in the proposed ddTASEP model, we hypothesize that the contribution from pausing induced by the slow nucleosome dynamics will dominate the first passage time eliminating the entrainment effect that is observed on bare DNA allowing us to coarse-grain the system and use a simpler, exponential waiting time distribution without introducing significant error. To demonstrate this, we first prove that the entry waiting time statistics are not affected by considering a larger number of intermediate entry steps. Second, we then demonstrate that the nucleosome induced pausing dominates the mean and the variance of the first passage time relative to the time to transcribe the bare DNA after entry into the nucleosome associated region.

First, we consider a more general transient state matrix $R'$ (analogous to Eq 17) which considers the entry of a polymerase of length $L$ that travels at rate $Lq$ (between subsequent base-pairs) into the nucleosome associated region that can be temporarily arrested by nucleosome rebinding at rate $h_c$ at any step of the process. The transient states of the new Markov Chain $R'$ will have a block structure with each two rows indicating an open (unarrested) and closed (arrested) state for each step of the extended polymerase's entry into the nucleosome associated region as given by

$$R' = \begin{bmatrix} -h_o & h_o & 0 & 0 & 0 & \cdots \\ h_c & -(h_c + Lq) & 0 & Lq & 0 & \cdots \\ 0 & 0 & -h_o & h_o & 0 & \cdots \\ 0 & 0 & h_c & -(h_c + Lq) & \ddots & \ddots \\ 0 & 0 & 0 & 0 & \ddots & \ddots \\ \vdots & \vdots & \vdots & \vdots & \ddots & \ddots \end{bmatrix} \tag{28}$$

As a representative example, we can consider the two-site case where the advance rate is $2q$. Interestingly, the nucleosome entry time for the two-site case $E_{e,2}$ was identical to that the of the one-site case ($E_{e,2} = E_e$). Additionally, the variance of the entry waiting time $V_{e,2}$ was found to be

$$V_{e,2} = \frac{(h_c + h_o)^2}{2h_o^2 q^2} + \frac{2h_c}{h_o^2 q} + \frac{h_c(h_c + 2h_o)}{h_o^2(h_c + h_o)^2} \tag{29}$$

Further, we expect that $V_{e,2} \approx V_e$. Evaluating the ratio of the two gives

$$\frac{V_{e,2}}{V_e} = \frac{(h_c + h_o)^4 + 4h_c(h_c + h_o)^2 + 2h_c(h_c + 2h_o)q^2}{2(h_c + h_o)^4 + 4h_c(h_c + h_o)^2 + 2h_c(h_c + 2h_o)q^2} \approx 1 \; for \; h_c, h_o \ll 1 \tag{30}$$

While analytical results become unwieldy as $L \to 50$, this approach can be performed numerically in MATLAB. The numerical values of $E_{e,50}$ and $V_{e,50}$ are plotted against the analytical results for $E_e$ and $V_e$ in S1 Fig for the same parameter sets shown in Fig 3. As hypothesized, S1A Fig shows that the expected entry waiting times are equal to each other. Likewise, S1B Fig shows that the variance of this waiting time for the fifty-step model is approximately equal to the variance for a single-entry step except for the data series when $h_c = 1$ (since the highest power term of $(h_c + h_o)$ can no longer vanish) and in cases where $h_c \ll h_o$, which correspond to effectively bare DNA (as demonstrated by the final data points of the $h_c = 0.001$ and $h_c = 0.01$ data series).

Next, in S1C Fig, we establish that the nucleosome entry time dominates the first passage time through a nucleosome associated region. From S1C Fig (left), for biologically interesting conditions when $h_o < h_c$, the bare DNA passage time contributes roughly 10% of the mean first passage time to clear a nucleosome in both models. With respect to the variance (S1C Fig, right), the bare DNA passage time becomes even less significant. In the exponentially distributed waiting time/single site case, the bare DNA passage time contributes less than 10% of the variance to the overall passage time (excluding cases with $h_o \gg h_c$ which constitute effectively bare DNA). While the single site model has a higher bare DNA contribution to the variance of first passage time relative to the fifty site model, the dominant source of variance (and thus pausing that disrupts the flux) is the nucleosome entry process for both cases.

Therefore, while a more accurate waiting time distribution would be critical in the absence of nucleosomes, the nucleosome induced pausing and subsequent polymerase-polymerase collisions within a burst group likely cancel out any of the entrainment effects that would be observed on bare DNA if a more realistic waiting time distribution were used. Thus, we believe that the use of a single site polymerase with 50 bp lattice sites and exponentially distributed waiting times is still an appropriate simplification to model the system.

## Constructing the ddTASEP $\alpha\gamma$-plane

The canonical TASEP phase diagram exhibits three jamming transitions in the $\alpha\beta$-plane when the initiation rate becomes limiting ($\alpha < \beta$ and $\alpha < \frac{1}{2}$), the termination rate becomes limiting ($\beta < \alpha$ and $\beta < \frac{1}{2}$), or when the channel capacity is reached in the max flux limit ($\alpha, \beta \geq 1/2$) as shown in Fig 1B. Analogously, the ddTASEP should exhibit a jamming transition due to the limitation on the maximum transcription flux imposed by the nucleosome-induced pausing. Therefore, we expected that the transcription efficiency $\gamma$ could be used to define a third axis of the phase diagram orthogonal to the canonical $\alpha\beta$-plane. By assuming $\beta = q = 1$, we restrict our analysis to the $\alpha\gamma$-plane as shown in Fig 4. The reader should note that, by the selection of $q = 1$, we are implicitly rescaling the values of $\alpha$ and $\gamma$ with respect to the advance rate $q$.

Classically, the ideal method to determine the value of the transcription observables $J$, $\rho$, and $\nu$ and to locate the phase transitions in parameter space would be to utilize a mean field approach [31,32]. However, this approach has significant limitations. Based on the model formulation in Eqs 1–6, it is clear that the system's chemical master equation would involve the products of up to four site occupancies. Therefore, when taking the ensemble average of the site occupancies, quadratic (e.g. $\langle s_i s_{i+1} \rangle$) and cubic mixed moments (e.g. $\langle s_{i+1} s_{i+2} s_{i+3} \rangle$) appear in the master equation. (Note that quartic moments such as $\langle h_m s_{i+1} s_{i+2} s_{i+3} \rangle$ vanish due to the exclusionary binding constraint.) Under the mean-field approximation (also referred to as the

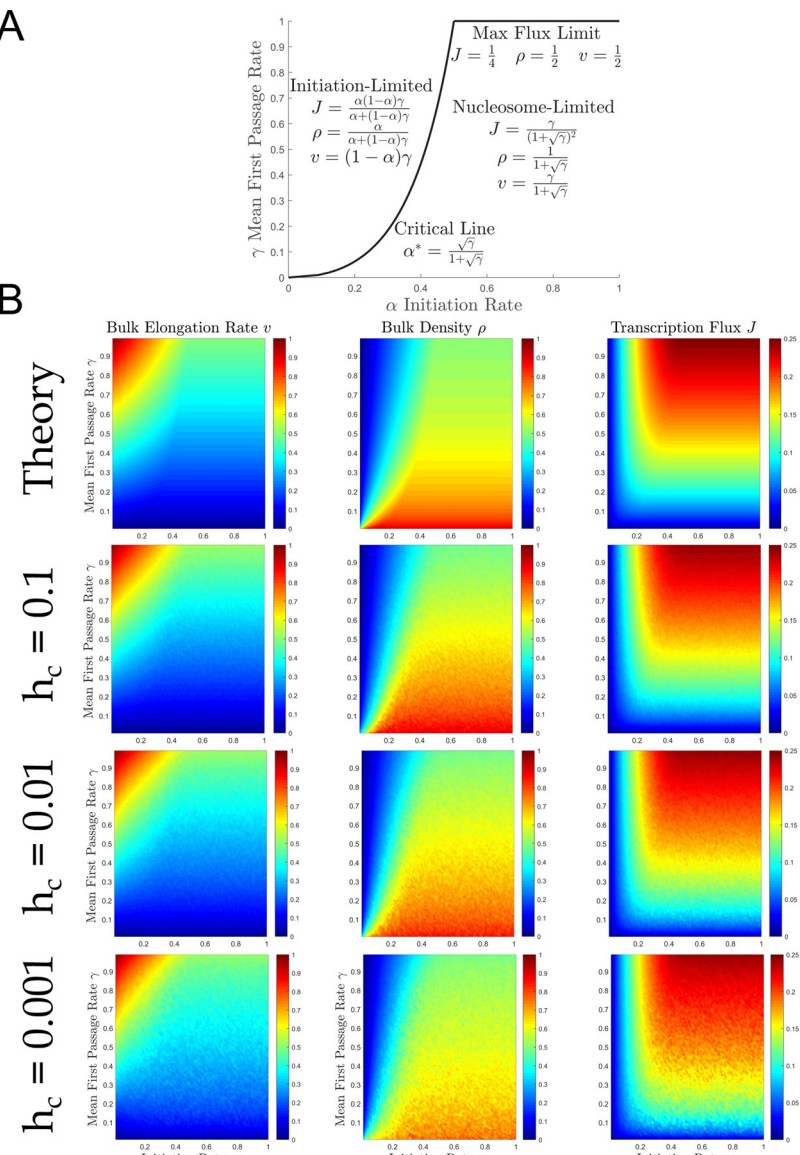

**Fig 4. The ddTASEP phase diagram in the αγ-plane.** (A) Phase diagram in the $\alpha\gamma$-plane (orthogonal to the original $\alpha$ß-plane in Fig 1B at ß = q = 1). The critical line (Eq 37) divides the phase diagram into two regions: the initiation-limited and the nucleosome-limited For $\gamma\to1$, the ddTASEP's dynamics converge to the classical Max Flux Limit shown in Fig 1B. (B) The theoretical predictions (top row) for the bulk elongation rate $v$ (Eq 38), bulk density $\rho$ (Eq 39), and bulk transcription flux $J$ (Eq 40) are qualitatively compared to simulated results for $h_c$ = 0.1, 0.01, and 0.001 (2nd through 4th rows) Mean first passage rate, $\gamma$ was varied from between 0 and 1 by specifying $h_c$ (for given $h_o$) to achieve the desired value. 1.2 million simulation steps were used for each heat map point.

random phase approximation for the TASEP), these mixed moments can be factored as

$$\langle s_i(1 - s_{i+1})\rangle \approx \langle s_i\rangle\langle 1 - s_{i+1}\rangle = \rho_i(1 - \rho_{i+1}) \tag{31}$$

which implies that each site of the classical TASEP comes to a steady state value and that no site occupancies are correlated with each other. The correlation-free assumption clearly fails for the proposed ddTASEP with extended dynamic defects and exclusionary defect binding as shown in Fig 2E and 2F. Therefore, we decided to abandon this approach entirely.

In order to derive approximate expressions for $J$, $\rho$, and $v$, we propose a phenomenological model inspired by saturation kinetics [54,55] that will interpolate between the canonical TASEP regime ($\gamma = 1$, $\alpha \in [0, 1]$) and the strongly perturbed ddTASEP regime with low initiation rates ($\gamma < 1$, $\alpha \ll 1$). Specifically, appropriate functional forms for $v$ and $\rho$ were selected to satisfy the assumed asymptotic behavior. The bulk elongation rate $v$ and the bulk density $\rho$ were then used to calculate $J$, and all three expressions were then compared against the simulated results.

First, we identified an appropriate functional form for the bulk polymerase hop rate $v$ in the regime with $\alpha < \frac{1}{2}$. In the canonical TASEP limit $\gamma \to 1$, the bulk hop rate $v$ is equal to $(1-\rho) = (1-\alpha)$ for $\alpha < \frac{1}{2}$ as shown in Eqs 1 and 2. Since the density in the initiation limited regime is $\rho = \alpha$ (Eq 2), the bulk hop rate $v$ must equal $(1-\alpha)$. As the initiation rate $\alpha \to 0$, the gene effectively resets to its resting state between each passing polymerase. Therefore, the bulk hop rate $v$ should approach $\gamma$. To satisfy both limits, we propose that

$$v := q\gamma(1 - \alpha) \; for \; \alpha < \frac{1}{2} \tag{32}$$

In order to estimate the bulk density $\rho$, we noticed that, in the limit $\gamma \to 1$, the bulk density $\rho$ must also converge to the classical TASEP bulk density in Eq 2. Additionally, in the limit $\gamma \to 0$ with a sufficiently large initiation rate $\alpha$, $\rho$ should be close to one in this limit since a whole gene spanning convoy may form (Fig 2B). Gene spanning convoys are able to form because the most likely binding site for nucleosomes is at the end of the gene (Fig 2D, top left), where the polymerase density drops to nearly zero. Last, in the highly perturbed cases, switching from an almost completely full gene to an almost empty gene is observed as shown in Fig 2B (top left). Under these conditions, the limit $\rho \to 1$ is not reached even though the gene may transiently be fully occupied.

From the canonical TASEP model, the density has a linear relationship with the initiation rate in the low-density region. For sufficiently high first passage elongation rates $\gamma$, the density should increase almost linearly with the initiation rate for small values of $\alpha$. However, the density will eventually saturate to some limit influenced by $\gamma$ since convoys will back up on the transcription start site preventing new initiation. We postulate that $\rho$ has a linear dependence on the initiation rate $\alpha$ (in the limit $\gamma \to 1$) and tends to one as $\gamma \to 0$. Therefore, we propose that

$$\rho := \frac{\alpha}{\alpha + (1 - \alpha)\gamma} \; for \; \alpha < \frac{1}{2} \tag{33}$$

Finally, we obtain the transcription flux $J$ as

$$J = \rho \times v = \frac{q\alpha(1 - \alpha)\gamma}{\alpha + (1 - \alpha)\gamma} \; for \; \alpha < \frac{1}{2} \tag{34}$$

In the limit $\gamma \to 1$, the value of $J$ converges to the unperturbed TASEP flux for the initiation limited regime (Eq 3) scaled by the advance rate $q$.

$$\lim_{\gamma \to 1} J = q\alpha(1 - \alpha) \; for \; \alpha < \frac{1}{2} \tag{35}$$

In the unperturbed TASEP, when $\alpha \geq \min\left(\frac{1}{2}, \beta\right)$, the flux (as well as the density and the hop rate) become independent of the injection rate, and the flux $J$ reaches its maximum value for a given $\beta$ equal to $\max\left[\frac{1}{4}, \beta(1 - \beta)\right]$. By analogy, it is reasonable to expect that, for a given $\gamma$, the flux of the ddTASEP will also become independent of the initiation rate $\alpha$ after some saturating condition is reached since the nucleosomes will cause the polymerases to back up onto

the TSS. We propose that for a given $\gamma$ the jamming transition will occur when $J(\alpha,\gamma)$ reaches its maximum value with respect the initiation rate $\alpha$, as given by

$$\frac{dJ}{d\alpha} = \frac{q\gamma[\gamma(1-\alpha)^2 - \alpha^2]}{[\alpha + (1-\alpha)\gamma]^2} = 0 \tag{36}$$

Therefore, the critical value of the initiation rate $\alpha^*$ is

$$\alpha^* = \frac{\sqrt{\gamma}}{1 + \sqrt{\gamma}} \tag{37}$$

For $\alpha < \alpha^*(\gamma)$, both $\alpha$ and $\gamma$ limit the rate of transcription. In the nucleosome-limited regime where $\alpha \geq \alpha^*(\gamma)$, the density, flux, and hop rate no longer depend on the initiation rate but now depend exclusively on nucleosome-induced pausing. Substituting $\alpha^*$ into Eqs 32, 33 and 34 gives the proposed estimates for $v$, $\rho$, and $J$.

$$v = \begin{cases} q\gamma(1-\alpha), \alpha < \alpha^* \\ v_{min}(\gamma) = \dfrac{q\gamma}{1 + \sqrt{\gamma}}, \alpha \geq \alpha^* \end{cases} \tag{38}$$

$$\rho = \begin{cases} \dfrac{\alpha}{\alpha + (1-\alpha)\gamma}, \alpha < \alpha^* \\ \rho_{max}(\gamma) = \dfrac{1}{1 + \sqrt{\gamma}}, \alpha \geq \alpha^* \end{cases} \tag{39}$$

$$J = \begin{cases} \dfrac{q\alpha(1-\alpha)\gamma}{\alpha + (1-\alpha)\gamma}, \alpha < \alpha^* \\ J_{max}(\gamma) = \dfrac{q\gamma}{(1 + \sqrt{\gamma})^2}, \alpha \geq \alpha^* \end{cases} \tag{40}$$

These results are summarized in the phase diagram for the $\alpha\gamma$-plane as shown in Fig 4A. The phase diagram is divided into initiation-limited and nucleosome-limited or jammed regions at the critical line defined Eq 37. The simulations shown in Fig 4B provide evidence that the analytical estimates (Eqs 38–40) reproduce qualitatively dependence of $v$, $\rho$, and $J$ on the model parameters.

### The initiation rate $\alpha$ and the mean first passage rate $\gamma$ control transcription dynamics

The results shown in Fig 4B suggest that Eqs 38–40 predict the transcriptional dynamics of the system regardless of the particular values of $h_o$ and $h_c$. This is confirmed by Fig 5A–5C which show the dependence of the bulk elongation rate $v$ (Fig 5A), bulk density $\rho$ (Fig 5B), and the transcription flux $J$ (Fig 5C) on the mean first passage rate $\gamma$ (which was obtained by varying $h_o$ while holding $h_c$ fixed across multiple orders of magnitude). The plots confirm that the values of $h_o$ and $h_c$ typically have relatively small effect on the transcription dynamics, which is nearly fully governed by $\gamma$.

The highest deviations exist for the lowest and highest values $h_c = 0.001$ (blue) and $h_c = 1$ (red). However, the predictions for the bulk elongation rate $v$ (Fig 5A), bulk density $\rho$ (Fig 5B), and transcription flux $J$ (Fig 5C) are the most accurate for the biologically relevant cases with $h_c = 0.01$ (green) and $h_c = 0.1$ (orange) across all three metrics. As a point of reference, if $q = 50\frac{bp}{s} = 1\frac{site}{s}$, then the nucleosome dynamics associated with

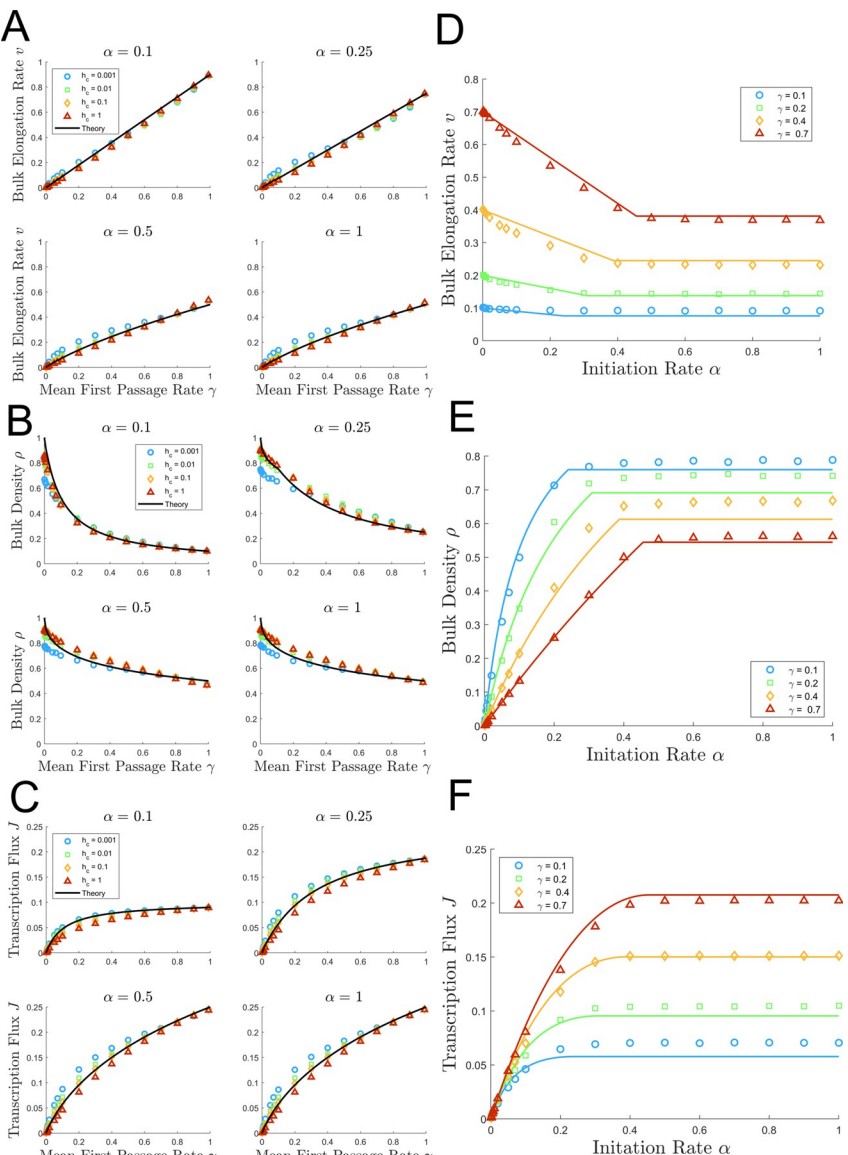

**Fig 5. Dependence of the elongation rate v, polymerase density ρ, and transcription flux J on the initiation rate α and the mean first passage rate γ.** The left panels A, B and C show the dependence of $v$ (Eq 38), $\rho$ (Eq 39), and $J$ (Eq 40) on $\gamma$. $\gamma$ is varied from 0.001 to 0.99 by adjusting $h_o$ for fixed values of $h_c$ equal to 0.001 (blue), 0.01 (green), 0.1 (orange), and 1 (red). For each subpanel of A, B, and C, a different $\alpha$ values is assumed: 0.1 (top left), 0.25 (top right), 0.5 (bottom left), 1 (bottom right). The right panels D, E and F show the dependence of $v$ (Eq 38), $\rho$ (Eq 39), and $J$ (Eq 40) on $\alpha$. $\alpha$ is varied from 0.001 to 1 for fixed values of $\gamma$ equal to 0.1 (blue), 0.2 (green), 0.4 (orange), and 0.7 (red) under the constraint that $h_o = h_c$. Data points represent the average simulation results from 50 simulations with $2 \times 10^6$ Monte Carlo steps each, and solid lines represent theoretical estimates.

$0.01 \leq h_c < 0.1\ s^{-1}$ would be on the order of 10–100 seconds, which is biologically relevant as shown by Bintu et al [13].

Fig 5D–5F show the dependence of the bulk elongation rate $v$ (Fig 5D), bulk density $\rho$ (Fig 5E), and transcription flux $J$ (Fig 5F) on $\alpha$. Because Fig 5A–5C show that $\alpha$ and $\gamma$ control the transcription dynamics with minimal effects from $h_o$ and $h_c$ under biologically relevant conditions, we decided to omit explicitly varying $h_c$ in Fig 5D–5F. However, the reader should note that $h_c = h_o$ that the nucleosome rate constants vary from 0.0143 to 0.7 to achieve values of $\gamma$ from 0.1 to

0.7 in Fig 5D–5F. The sharp jamming transition that arises when initiation rate passes the critical value $\alpha^*(\gamma)$ is readily visible in these figures. Fig 5F shows that, as predicted by the model, the transcription flux increases linearly with $\alpha$ until $\alpha$ reaches the critical value $\alpha^*(\gamma)$ at which $J(\alpha,\gamma)$ reaches its maximum value $J_{max}(\gamma)$ (Eq 40). Similarly, the bulk density $\rho(\alpha,\gamma)$ reaches its maximum value $\rho_{max}(\gamma)$ (Eq 39) in Fig 5E while the bulk elongation rate $v(\alpha,\gamma)$ reaches its minimum value $v_{min}(\gamma)$ (Eq 38) in Fig 5D. It is worth nothing that our theoretical predictions slightly overestimate the bulk elongation rate and slightly underestimate the bulk density. However, since the transcription flux is the product of these two quantities, the errors nearly cancel out leading to quantitative agreement between the simulated results and theoretical estimate for $J(\alpha,\gamma)$ as shown in Fig 5F. For $J(\alpha,\gamma)$ the transition between the initiation-limited and nucleosome-limited regimes is smooth (in contrast to the sharp transitions for $\rho$ and $v$ in agreement with assumption that $J(\alpha, \gamma)$ has zero derivative with respect to $\alpha$ at $\alpha^*(\gamma)$ (Eq 37).

## ddTASEP transcription dynamics are not affected by gene length or geometry for short convoy lengths

Fig 6 investigates the effects of varying gene length and the length of linker regions between nucleosomes. Fig 6A–6C show the dependence of elongation rate, bulk density, and transcription flux on the length of the gene, ranging from 2,000 to 20,000 base pairs. On short genes, the convoys start to span the length of the gene, giving rise to some interesting effects. Most notably, the observed transcription flux is higher than that predicted by our proposed model, and the observed density is lower than our proposed model on short genes. We hypothesize that this is connected to the existence of gene spanning convoys. When convoys span the length of the gene, nucleosomes can only infrequently re-bind, and the gene transiently approximates classical TASEP behavior. In the initiation limited regime, the transcription flux of a classical TASEP is the upper-bound of transcription flux for all other related ddTASEPs (with $\gamma < 1$), and the density of a classical TASEP is the lower-bound of density for all other analogous ddTASEPs. Therefore, the previously mentioned elevated transcription flux and decreased bulk density are consistent with more classical TASEP behavior that may occur in a gene-spanning convoy. Accordingly, when the gene length increased to over 10,000 base pairs, the transcription flux, density, and elongation rate converge to the theoretical predictions for the ddTASEP since the formation of a gene-spanning convoy (that would induce quasi-classical TASEP dynamics) becomes too unlikely for the given rate of nucleosome dynamics.

Fig 6D–6F show the dependence of the bulk ddTASEP properties on the linker spacing between nucleosomes on a 20,000 base pair gene. The reader should note that the number of nucleosomes decrease in the simulation as the number of linker sites increase. Even when $\gamma$ must be calculated assuming different linker geometries, the analytical results for the bulk elongation rate (Fig 6D) and transcription flux (Fig 6F) adapt almost perfectly to the new geometry and perform reasonably well for the bulk density (Fig 6E).

## Features of nucleosome-induced convoy formation and transcriptional bursting

While classical models of transcription attribute bursting to promoter dynamics where the promoter switches between an ON and OFF state [26,27,56–59], the ddTASEP model exhibits bursting even with a constitutively active promoter because of nucleosome-induced pausing. In order to identify bursts, we utilized the difference in time scales between the average intra-convoy waiting time $\overline{t_{in}}$ and the average inter-burst interval $\overline{t_{IBI}}$. As can be seen in Fig 2A and 2B, nucleosome induced pausing introduces non-trivial time delays between the last polymerase of a convoy and the pioneering polymerase of the convoy following it. To briefly

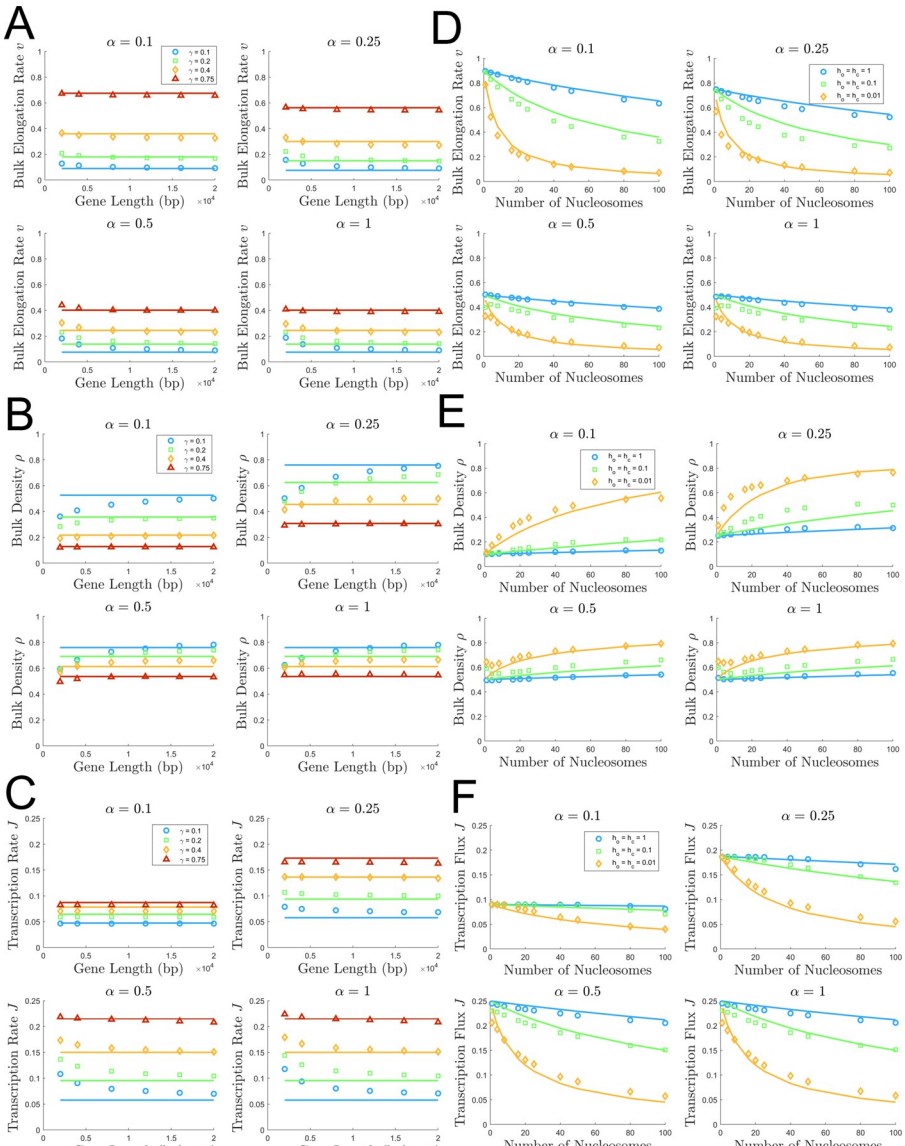

**Fig 6. Evaluation of the effects of gene length and nucleosome spacing on bulk transcriptional properties.** Panels A ($v$), B ($\rho$), and C ($J$) show the results of a parameter sweep with respect to gene length from 2,000 to 20,000 bp with $\alpha$ = 0.1 (top left), 0.25 (top right), 0.5 (bottom left), and 1 (bottom right), and $\gamma$ = 0.1 (blue), 0.2, (green), 0.4 (orange), and 0.75 (red). Panels D ($v$), E ($\rho$), and F ($J$) show the results of a parameter sweep with respect to the number of evenly spaced nucleosomes (by varying number of linker sites) giving $N_{Nuc} \in \{1, 4, 8, 16, 20, 25, 40, 50, 80, 100\}$ with $\alpha$ = 0.1 (top left), 0.25 (top right), 0.5 (bottom left), and 1 (bottom right). Since $\gamma$ depends on geometry, $h_o$ and $h_c$ were set equal to 1 (blue), 0.1 (green), and 0.01 (orange). Both sweeps were performed with $2 \times 10^6$ Monte Carlo steps and 50 replicates.

summarize our approach (described in full in the methods), we utilized a kernel density estimate of $\log_{10} t_w$ weighted by itself to determine whether the system exhibited bursting and (in the presence of bursts) to identify the approximate location of the average inter- and intra-burst intervals. The threshold for a burst waiting time was set as the geometric average of the two peaks of the kernel density estimate. Representative kernel density estimates for the parameter sets used in Fig 2 are shown in Fig 7A. Excluding the cases without strong

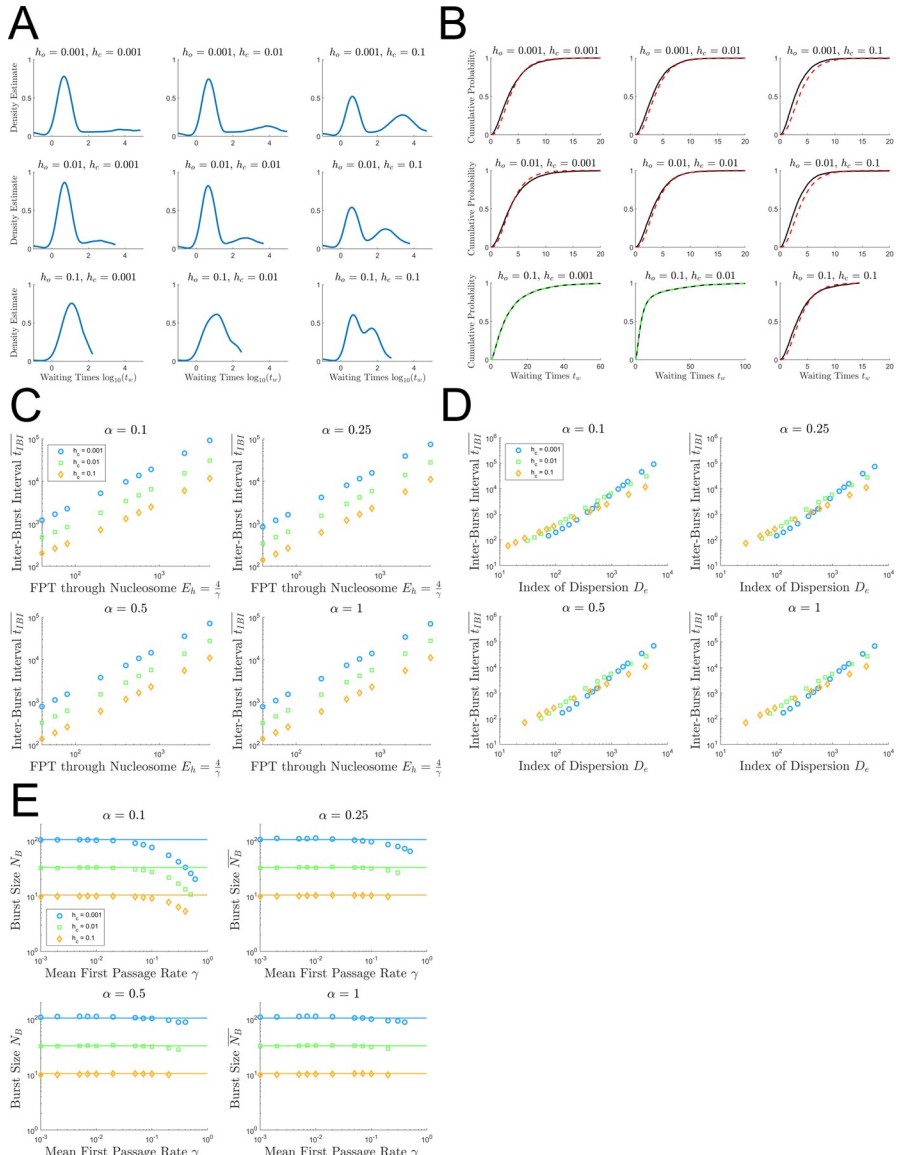

**Fig 7. Investigation of the ddTASEP Inter-Burst Interval and Average Burst Size.** (A) Representative smoothed kernel density estimates of $\log_{10}(t_w)$ weighted by $\log_{10}(t_w)$ using same parameter sets as Fig 2. (B) Empirical cumulative distribution functions of waiting times (black line) from ddTASEPs with the same parameters as in Fig 2. The time-headway cumulative distribution function is overlaid for each plot (Eq 42). The red dashed time headway distributions have robust bursting with an effective density $\rho_{eff} = 0.5$. In contrast, the green dashed lines indicate initiation limited cases where unimodality was observed in Fig 7A, and $\rho$ is given by Eq 39. (C) shows the average inter-burst interval $\overline{t_{IBI}}$ plotted as a function of $E_h = 4/\gamma$. (D) shows the average inter-burst interval $\overline{t_{IBI}}$ plotted as a function of index of dispersion of the first passage time to enter a nucleosome $D_e$. (E) shows burst size $\overline{N_B}$ plotted as a function of mean first passage rate $\gamma$ with the solid lines proportional to $\frac{1}{\sqrt{h_c}}$. Panels (C), (D), and (E) were generated with Fig 5A–5C. $\gamma$ is varied from 0.001 to 0.99 by adjusting $h_o$ for fixed values of $h_c$ equal to 0.001 (blue), 0.01 (green), and 0.1 (orange) omitting cases that failed to show bimodality.

correlations (Fig 2F, bottom left and bottom middle), all of the remaining cases were strongly bimodal indicating that non-trivial bursting is occurring.

Next, we investigated the internal transcription dynamics of bursting ddTASEPs versus non-bursting ddTASEPs in Fig 7B. In the classic TASEP, the waiting time between particles is

given by the time headway distribution $f(t;\rho)$. The probability density function $f(t;\rho)$ of this distribution is given by [60]

$$f(t;\rho) = \frac{\rho}{1-\rho}\left[e^{(1-\rho)t} - 1\right]e^{-t} + \frac{1-\rho}{\rho}\left[e^{\rho t} - 1\right]e^{-t} - te^{-t} \qquad (41)$$

and its cumulative distribution function $F(t;\rho)$ is given by

$$F(t;\rho) = \frac{1}{1-\rho}(1 - e^{-\rho t}) - \frac{\rho}{1-\rho}(1 - e^{-t}) + \frac{1}{\rho}\left(1 - e^{-(1-\rho)t}\right) - \frac{1-\rho}{\rho}(1 - e^{-t}) + e^{-t}(1 + t) - 1 \,(42)$$

The mean of this distribution is known to be $\frac{1}{\rho(1-\rho)} = \frac{1}{J} = 4$. For a burst group at the end of a gene, we hypothesized that it would exhibit max flux limit TASEP dynamics since the entry rate into the burst group is given by $q$ and the termination rate is given by $\beta = q$. Therefore, the effective density of a burst group would be $\rho = 0.5$. In contrast, for a case without bursting, the density of the system would be given by our initiation limited estimate in Eq 39. The empirical cumulative distribution functions in Fig 7B directly confirm this result.

In Fig 7C, the relationship between the average inter-burst interval $\overline{t_{IBI}}$ and the first passage time to clear a nucleosome $E_h = 4/\gamma$ was considered. While there is a clear correlation between $E_h$ and $\overline{t_{IBI}}$ ($r = 0.68$), there is clearly dependence on the magnitude of $h_c$ that is not appropriately accounted for here. Therefore, we investigated the hypothesis that the inter-burst interval should scale with index of dispersion of the time to enter a nucleosome since $D_e$ quantifies the deviation from classical TASEP behavior. The variance $V_e$ will be directly related to the observed inter-burst intervals $t_{IBI}$ since these contribute most strongly to the variance out of all of the observed waiting times $t_w$ since the intra-burst intervals are always approximately $\overline{t_{in}} \approx 4$. However, a system with a long MFPT with frequent, short inter-burst intervals $t_{IBI}$ and a system with a short MFPT with infrequent, long inter-burst intervals $t_{IBI}$ could have comparable variances. Therefore, to better account for the relative perturbation strength and to maintain dimensional consistency, the index of dispersion $D_e$ is preferable to the variance $V_e$. Fig 7D shows that all three data series collapse onto a single curve and that the average inter-burst intervals exhibit a power law scaling with $D_e$ that could be slightly improved with additional correction for $\alpha$ and $h_c$. However, the correlation is still strong with $r = 0.82$ even though the relationship is not completely linear.

Fig 7E investigates the limiting values of burst size. Under saturation conditions, the maximum average burst size is approximately

$$\overline{N_{B,max}} \approx \frac{3.3}{\sqrt{h_c}}, for\ \alpha \geq \alpha^*(\gamma) \qquad (43)$$

This result is surprising since it initially seems like burst size should be controlled by the flux and the absolute time scale of the nucleosome unwrapping ($1/h_o$) because this should be connected to the number of polymerases that can accumulate behind a closed nucleosome. However, when the gene is saturated, a constant number of polymerases accumulate behind each closed nucleosome. The limiting factor under these conditions is how fast the nucleosome can close in such a way that it splits burst groups to finite size as shown in the anti-correlated regions observed in the Fig 2F.

## The existence of bursts under initiation limited conditions

While robust bursting is clearly evident in the nucleosome-limited (saturated) region, it is unintuitive whether or not robust bursting occurs in the initiation limited region. In Fig 8A, we investigate the average inter-burst interval $\overline{t_{IBI}}$ as $\alpha{\to}0$. The predictions given by the time

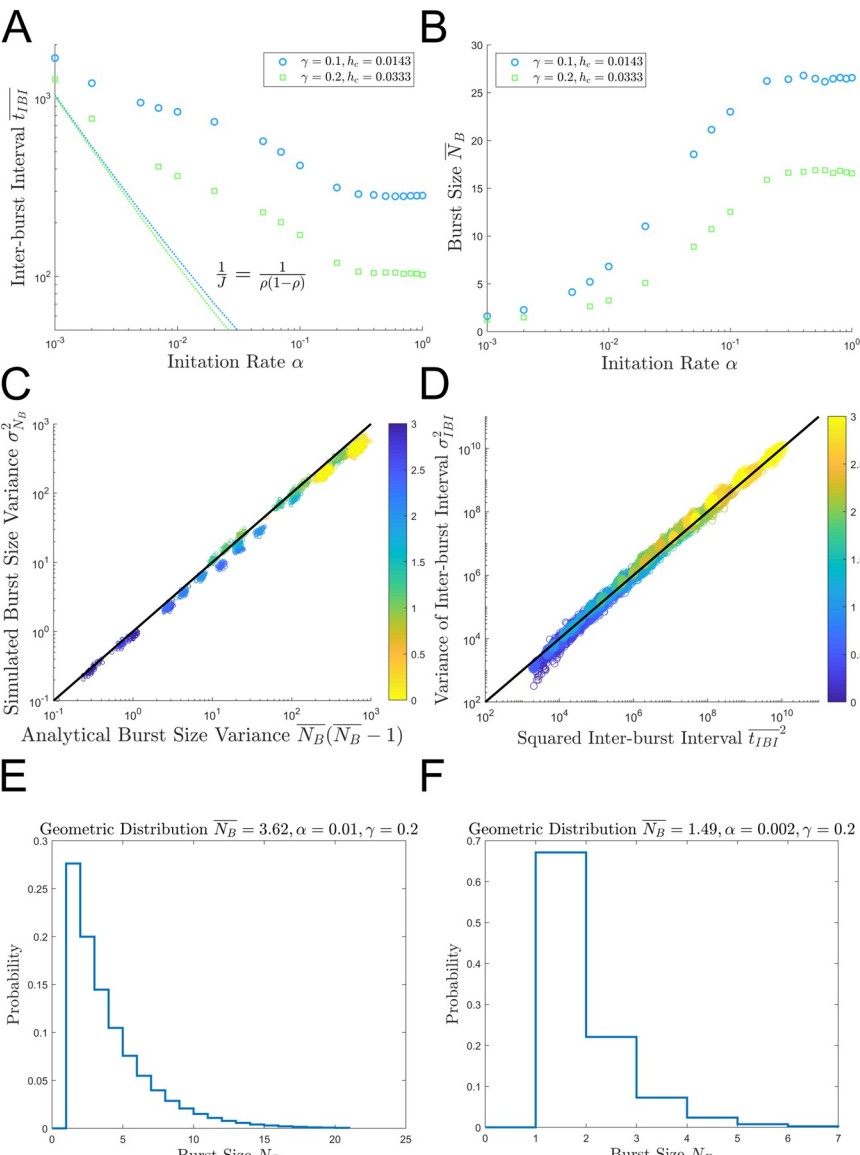

**Fig 8. Bursting in the low initiation rate regime.** (A) shows the average inter-burst interval $\overline{t_{IBI}}$ as a function of $\alpha$ with the headway distribution prediction for the average waiting time $\overline{t_w}$ given by a dashed line. (B) shows the average burst size $\overline{N_B}$ plotted as a function of $\alpha$. Panels (A) and (B) were generated with Fig 5D–5F with $\alpha \in [0.001, 1]$ with $h_c = h_o \in \{0.0143, 0.033\}$ to achieve $\gamma \in \{0.1, 0.2\}$. (C) shows that the variance of burst size $\sigma^2_{N_B}$ (simulated) is equal to the variance of a geometric distribution given by Eq 46 and is colored by $-\log_{10} \alpha$. (D) compares the mean inter-burst interval and the variance of the inter-burst intervals to establish that the waiting time distribution is exponentially distributed using the data from Fig 5A–5C and is colored by $-\log_{10} \gamma$. (E) and (F) show representative burst size probability mass functions generated from geometric distributions with means given by $\overline{N_B} = 3.62$ and 1.49 associated with the parameter sets ($\alpha = 0.01, \gamma = 0.2$) and ($\alpha = 0.002, \gamma = 0.2$).

headway distributions ($1/J$) are given by the dashed lines. Remarkably, the inter-burst intervals are substantially longer than the waiting times between randomly initiated polymerases. In other words, the $\overline{t_{IBI}}$ now represents the sum of the waiting time between rare initiation events plus the sum of the time to clear the nucleosomes which have all returned to resting nucleosome occupancy since the polymerase density is approaching zero. Fig 8B directly confirms

the existence of burst groups. In the limit $\alpha \to 1$, the max burst size converges to the value given by Eq 43. In contrast, in the limit as $\alpha \to 0$, the burst size approaches one. However, our proposed burst identification procedure still identifies bursts for these parameter sets. We hypothesize that the burst sizes are geometrically distributed and that most "apparent bursts" under extreme initiation limited conditions are single polymerases while a smaller subset constitute non-trivial polymerase convoys/burst events.

We hypothesize that the burst sizes are geometrically distributed [27] such that

$$P(N_B = k) = (1-p)^{k-1}p \tag{44}$$

where $k$ is the number of polymerases in the burst, and $p$ is defined by

$$p = 1/\overline{N_B} \tag{45}$$

The variance of the burst size geometric distribution $\sigma^2_{N_B}$ is given by

$$\sigma^2_{N_B} = \frac{1-p}{p^2} = \overline{N_B}(\overline{N_B} - 1) \tag{46}$$

Fig 8C verifies that this relationship holds for the data set plotted in Fig 8A and 8B, so the assumption that the burst sizes are geometrically distributed is reasonable. Additionally, Fig 8D shows that the variance of the inter-burst intervals is equal to their mean squared, $(\sigma^2_{IBI} = \overline{t_{IBI}}^2)$ indicating that burst waiting times are exponentially distributed.

Interestingly, these findings bare some similarity to that of a two-state random telegraph model. In this classical model, mRNA are produced continuously for brief periods of time while the promoter of a gene is active, and then, mRNA production stops when the promoter is inactivated [61,62]. This model can be interpreted through the framework of the classical TASEP (without nucleosomes) by considering the association and dissociation of a transcription factor with rates $f_a$ and $f_d$ respectively. In this new framework, the promoter would be active for a time period of $1/f_d$ (i.e. when the transcription factor dissociates) and the inter-burst interval would be exponentially distributed with mean $1/f_a$ (i.e. the waiting time for transcription factor association). Burst groups would move deterministically along the length of the gene with the same bulk velocity, but the density and flux would be scaled by the fraction of time the promoter was active $\left(\frac{f_a}{f_a+f_d}\right)$. The burst size distribution would then be geometrically distributed with a burst termination probability of $\frac{f_d}{J+f_d}$ and a mean of $\frac{J}{f_d}$ (i.e. the flux times the expected time the promoter will remain active).

As in the two-state telegraph model, the burst sizes of our proposed ddTASEP also follow a geometric distribution, and the inter-burst intervals are exponentially distributed [27]. However, there are a few key distinctions. First, the burst sizes in our proposed model have an upper bound given by Eq 43 that is proportional to $1/\sqrt{h_c}$ and is completely independent of the flux. Second, if $J<f_d$ in a two-state telegraph model, polymerases will never form burst groups since the average number of polymerases initiated per active period is less than one. In contrast, in the ddTASEP, polymerases merge into convoys along the length of the gene due to repeated nucleosome induced pausing and arrive in rapid succession with waiting times distributed according to the time headway distribution of a classical TASEP in rapid succession, even for extremely small initiation rates.

Last, we will verify the ddTASEP's capacity to induce bursting for two cases in the extreme initiation limited regime for the parameter sets ($\alpha = 0.01$, $\gamma = 0.2$) and ($\alpha = 0.002$, $\gamma = 0.2$) which produced average burst sizes $\overline{N_B}$ of 3.62 and 1.49, respectively. The probability mass functions given by Eq 44 are plotted for both cases in Fig 8D and 8E. For the case with the

higher initiation rate ($\alpha = 0.01$) in Fig 8D, bursts of up to 15 polymerases can be observed in some rare events. More interestingly for the case with ($\alpha = 0.002$) in Fig 8E, the convoys can regularly include up to six polymerases even though the bulk density $\rho$ for this case is 0.01 and the flux $J$ is 0.002 (which would correspond to a 500 second average waiting time in a classical TASEP). In short, non-trivial bursting is still possible in the extreme initiation limited region.

From a biological standpoint, these parameter sets are obtainable. In Fig 8, cases with $\gamma$ equal to 0.1 and 0.2 are considered. Veloso et al. utilized BruDRB-seq to demonstrate that the range of first passage elongation rates $\gamma$ in K562 Leukemia cells ranged from 0.0083 to 1 site/s [19]. Bintu et al. showed via optical trap experiments that nucleosome unwrapping took between 10–100 seconds [13], and the cases presented here assume $h_o = h_c = 0.0143$ or 0.0333 which correspond to roughly 30–70 seconds. Tantale et al.'s study of HIV1 gene transcription (under high initiation rates) via MS2-tagging suggested that polymerases could move in convoys of 10–25 with an intra-burst waiting time of 4.3 seconds which is consistent with the predictions from the proposed geometric distribution for burst size and the TASEP time headway distribution [63]. Therefore, nucleosome-induced pausing serves a viable mechanism for transcriptional bursting on its own.

## Comparing the biological effects of static and dynamic defects

In this section, we numerically analyze RNAPII elongation on non-uniform DNA where the advance rate $q$ or the nucleosome rate constants $h_o$ and $h_c$ may vary along the gene. We assume that the gene contains a 1600 bp (8 nucleosomes/32 site), partially methylated CpG Island at the promoter and seven 400 bp (2 nucleosomes/8 site) regions that correspond to exons and the adjacent intron-exon boundary regions where co-transcriptional splicing occurs [18,64]. (The reader should note most genes contain one long exon of median length 1,000–1,500 bp and many smaller exons of median length 150–200 bp [65]).

For the cases where we treated these features as static defects, we assigned a unique value of $q_i$ to each site $i$. First, the effect of variable GC content was incorporated into each site's advance rate by the following equation

$$q_i = q \frac{[0.5N_{GC} + 1.5(50 - N_{GC})]}{50} \text{ where } N_{GC} \sim Binom(N = 50, p = 0.5) \qquad (47)$$

where the average advance rate is set by $q$, and $N_{GC}$ represents a binomial-distributed sample of the number of guanine and cytosine base-pairs present in the 50 bp interval. Then, the values of $q_i$ (obtained from Eq 47) were reduced by an additional 50% on the sites corresponding to the CpG Island and the exons. The minimum advancement rate obtained from this procedure (i.e. static defect slowdown of $q_i = 0.5q$ plus additional GC content penalty) used to generate the lattice for the gene in Fig 9A and 9B was $q_s = 0.43$.

For the cases where the pause inducing features were assumed to modulate the nucleosome dynamics, we assigned a unique pair of rate constants $h_{c,m}$ and $h_{o,m}$ to the $m^{th}$ nucleosome. The effect of variable GC content was incorporated into each pair of rate constants as follows

$$h_{c,m} = h_c \frac{[1.25N_{GC} + 0.75(150 - N_{GC})]}{150} \text{ where } N_{GC} \sim Binom(N = 150, p = 0.5) \qquad (48)$$

$$h_{o,m} = h_o \frac{[0.75N_{GC} + 1.25(150 - N_{GC})]}{150} \text{ where } N_{GC} \sim Binom(N = 150, p = 0.5) \qquad (49)$$

where $h_c$ and $h_o$ represent the average wrapping and unwrapping rate and $N_{GC}$ now considers 150 base pairs instead of 50 to reflect the size of the nucleosome associated region. On the CpG island and on the exons, the values of $h_{c,m}$ were increased by an additional 25% while the values

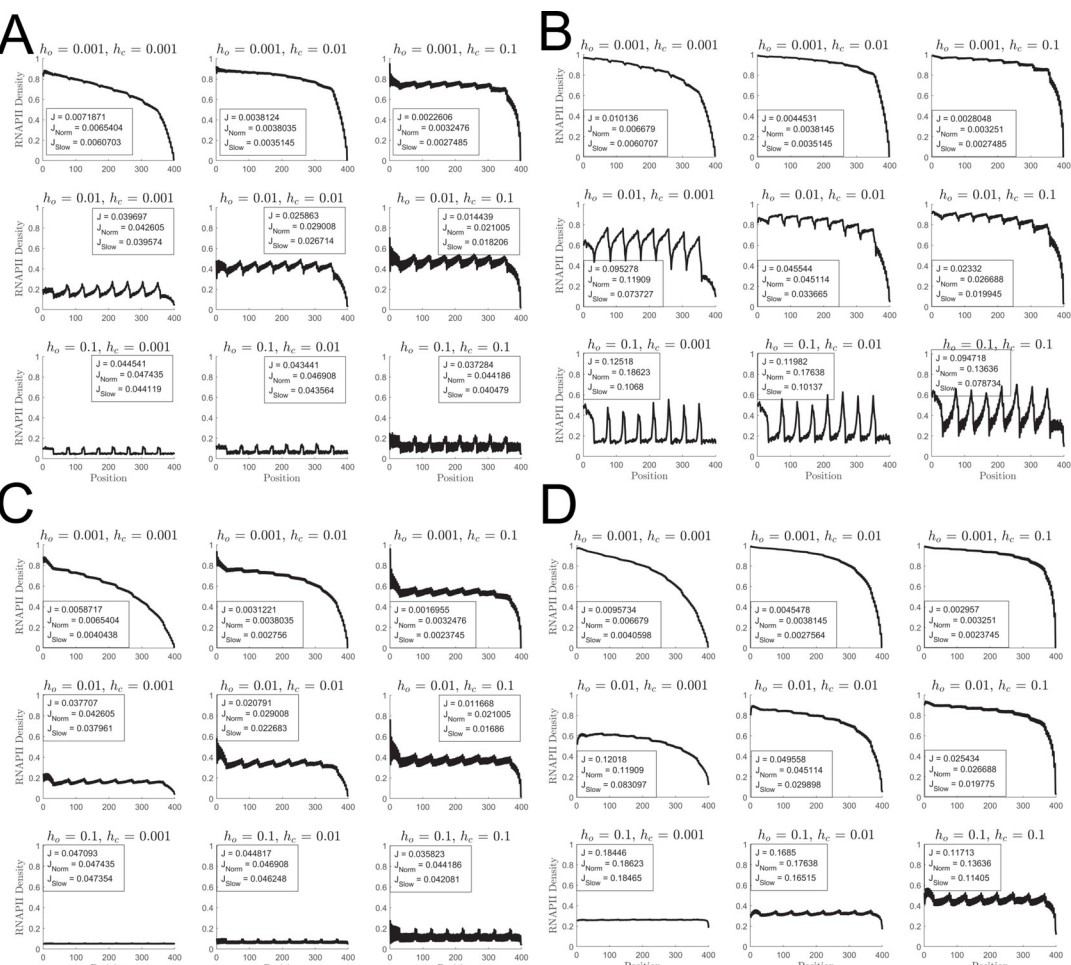

**Fig 9. Effects of static defects and nucleosome rate constant variability on RNAPII density profiles.** All subpanels have $h_c$ increasing from 0.001 to 0.1 from left to right and $h_o$ increasing from 0.001 to 1 from top to bottom as in Fig 2. 50 replicates of $N_{MCS} = 2\times10^6$ Monte Carlo steps are performed for all cases. All plots list the simulated flux $J$, the theory prediction for the unperturbed system $J_{Norm}$, and the theory prediction based on the slowest site $J_{slow}$. The left column (A, C) contains simulations at a low initiation rate of $\alpha = 0.05$ while the right column (B, D) contains simulations at $\alpha = 0.25$ which is saturating for the slow sites. The top row (A, B) shows density profiles from a gene with static defects representing a CpG Island (32 sites) on the promoter and 7 exons (8 sites) with a static defect advance rate of $q_i = 0.5q$ on these sites with additional GC content heterogeneity given by Eq 48 leading to a minimum advance rate $q_s = 0.43$. The bottom row (C, D) shows density profiles from the same gene, but the variability is introduced through the rate constants $h_{o,m}$ and $h_{c,m}$. GC content heterogeneity is introduced via Eqs 49 and 50. On the CpG Island and exon sites, $h_{c,m}$ is increased by 25%, and $h_{o,m}$ is decreased by 25%. With the addition of GC content heterogeneity, the maximum observed max($h_{c,m}$) was 27% higher than the average, and the minimum observed min($h_{o,m}$) was 26% lower than the average.

of $h_{o,m}$ were decreased an additional 25% from the values obtained from Eqs 48 and 49. Our random sampling procedure (i.e. $h_{c,m} = 1.25h_m$ + additional GC content enhancement and $h_{o,m}$ + GC content penalty) used to generate the gene in Fig 9C and 9D led to the max close rate being 27% higher than the average $h_c$ and the minimum open rate being 26% lower than the average $h_o$.

In Fig 9A and 9B, the effects of static defects on advance rate parameter variability is considered under initiation limited ($\alpha = 0.05$, 9A) and saturating conditions ($\alpha = 0.25$, 9B). The reader should note that all of our prior definitions for $v$, $\rho$, and $J$ given in Eqs 39–41 assume that $\alpha$ and $\gamma$ are scaled relative to $q$. Therefore, while $\alpha = 0.25$ is not saturating for $q = 1$, the rescaled $\alpha' = \frac{\alpha}{q_s} = \frac{0.25}{0.43} = 0.58$ is saturating for the slow sites. A few features become

immediately obvious. First, regions of elevated polymerase density have appeared at the sites of the static defects due to the increased dwell time on these sites, and the reduced flux through these sites has non-trivially reduced the transcription flux through the system (which must be constant across all sites). As the nucleosome dynamics slow down, the localized regions of elevated polymerase density smear out until the density profiles begin to qualitatively look like those in Fig 2C in the absence of perturbations.

In Fig 9A (barring the top left two plots), the flux through the gene is roughly equal to or slightly less than the flux predicted by the slowest bottleneck. The most significant discrepancies arise for cases with $h_c \gg h_o$ (top and middle right) where the simulated flux is significantly lower than the predicted flux through the slow sites. We hypothesize that this results from a synergistic effect where slow downs on the static defects increase the likelihood of nucleosome rebinding immediately after them. In contrast, for Fig 9B (barring the top left two cases), the fluxes mostly lie between the upper and lower bounds set by the unperturbed and most perturbed predictions. However, in the case of fast nucleosome dynamics in Fig 9B (bottom row), the simulated values converge to the predictions for the slowest site. Last, the discrepancies between the predicted and simulated fluxes that existed for the cases with $h_c \gg h_o$ when $\alpha =$ 0.05 (Fig 9A, top and middle left) have disappeared when the initiation rate was increased to $\alpha$ = 0.25. We suspect that saturating the gene has led to the formation of robust convoys that overcome the nucleosome rebinding effect that occurs after slow sites.

While Fig 9A and 9B demonstrate that static defects can throttle the flux, the resulting density profiles in Fig 9B have sharp discontinuities. Next, in Fig 9C and 9D, we investigate the alternative hypothesis that these genetic and epigenetic features could be inducing pausing by causing local variation in the nucleosome rate constants without inducing the same discontinuities.

In Fig 9C and 9D, it is clear that this alternative mechanism of action produces a similar throttling effect but with a smoother density profile. In Fig 9C, the simulated flux is dramatically reduced even relative to the fluxes in the analogous subplots in Fig 9A suggesting that small variations in the nucleosome rate constants can have a substantially stronger effect than even adjusting the nominal advance rate. Second, in many cases, the observed flux is substantially lower than the prediction for slow sites. We suspect that this effect is associated with the sharp decay in density at the left-hand boundaries in Fig 9C that is not observed in the static defect case. We hypothesize that the faster nucleosome rebinding on the promoter disrupts the initial formation of convoys until later on the gene causing a reduction in flux due to increased polymerase-nucleosome collisions.

While Fig 9C shows that local modification of the nucleosome dynamics can affect the flux at low initiation rates, Fig 9D shows the limitations of this hypothesis at higher initiation rates. For higher initiation rates, the exclusionary binding rule leads to nucleosome unwrapping/ destabilization even on the faster binding sites. Thus, when the system starts to saturate, the observed fluxes increase from at or slightly below the lower bound set by $J_{Slow}$ to an intermediate value between the bounds or even the upper bound set by $J_{Norm}$.

## Discussion

In this study, we present a stochastic, agent-based model of transcription with nucleosome induced pausing that maps onto the ddTASEP. The model reproduces key features of transcription including cooperative nucleosome destabilization [23,24,66], polymerase convoy formation [63], and transcriptional bursting [56]. Further, the model provides significant insight into the physics of jamming transitions in the presence of dynamic defects [34].

In lieu of a traditional mean-field approach to solving the ddTASEP dynamics, we calculated the moments of first passage time (MFPT and VFPT) using a Markov chain approach

[37,39], yielding exact results for both. Our approach allowed us to directly quantify the effects of the microscopic nucleosome rate constants on the first passage elongation rate $\gamma$. Through our analysis of the index of dispersion of the waiting time to enter a nucleosome ($D_e$), we observed that significant deviation from classical TASEP behavior can occur as the resting nucleosome density $\frac{h_c}{h_o+h_c}$ increases, but these deviations from classical behavior are more dramatic as the unwrapping kinetics slow down ($h_o \rightarrow 0$) even at fixed nucleosome density. This is potentially consistent with observations such as those in Bintu et al. that showed that multiple mechanisms to reduce transcriptional pausing exist. In acetylation, the thermodynamic equilibrium of the resting nucleosome density (wrapping state) is altered with minimal effect on the rates of wrapping and unwrapping [13]. In contrast, tail-less nucleosomes (which are a minimally explored epigenetic regulatory mechanism [67]) had substantially faster rates of unwrapping and re-wrapping with no change in the resting nucleosome density (wrapping state) [13]. Both mechanisms contributed to fewer transcriptional pauses of shorter duration relative to transcription through unmodified nucleosomes.

Using the mean first passage elongation rate $\gamma$, we constructed a new axis to the fundamental TASEP $\alpha\beta$-phase diagram. Subsequently, we developed analytical approximations for the core transcriptional properties (transcription flux $J$, bulk density $\rho$, and bulk elongation rate $v$) based on assumptions about the asymptotic behavior of the system for low initiation rates and near the max flux limit. We identified a novel jamming transition in the $\alpha\gamma$-plane that separated the transcriptional dynamics into initiation limited and nucleosome (dynamic defect) limited regions. Additionally, these estimates were robust to changes in gene length and to changes in the geometry of nucleosome spacing. Further research is merited into the physical properties of this system. For instance, the effects of varying $\beta$ in the $\alpha\beta\gamma$-volume and the effects of varying the dynamic defect length should be explored.

While the burst size and waiting time distributions shared similarities with two-state models, the model provided insight into a novel mechanism for transcriptional bursting that does not intrinsically rely on standard mechanisms at the promoter (such as two-state models) [25,26,56]. ddTASEP burst groups were shown to be quantitatively (via the time headway distribution [60]) similar to shorter TASEPs nested within the ddTASEP lattice. Further, the average inter-burst intervals were shown to correlate strongly with the index of dispersion $D_e$, and the max burst size was observed to be proportional to $\overline{N_B} \propto 1/\sqrt{h_c}$. In eukaryotic systems, many transcriptional initiation rates are slow with most genes showing one RNAPII on a loci on average at a time [68]. However, even in the extreme low initiation rate region, non-trivial transcriptional bursts were observed on average. Further, given that the burst sizes were geometrically distributed, non-trivial bursting can occur even in extreme initiation limited region.

While the nucleosome-mediated bursting hypothesis is interesting, our model more importantly demonstrates both how powerfully and how versatilely nucleosomes dynamics can modulate transcriptional elongation and flux. An emerging body of evidence is forming in the literature that implicates the role of enzymes such as BRD4 and DNMT1/DNMT3B as epigenetic drivers of cancer and epithelial to mesenchymal transition. BRD4 is a histone acetyltransferase associated with histone displacement, chromatin de-compactification, and cell cycle progression. BRD4's upregulation has been associated with EMT in hepatocellular carcinoma and in salivary adenoid cystic carcinoma [69–71]. DNMT's are DNA methyltransferases that maintain gene silencing by hypermethylating CpG islands which encourages tighter nucleosome occupancy. Loss of DNMT activity leads to loss of nucleosomes on the promoter of tumor suppressor genes leading to their transcriptional activation [72]. The phase diagrams of the proposed ddTASEP confirm that even minor changes of the nucleosome rate constants will significantly adjust $\gamma$ giving rise to a large dynamic range of behavior even for fixed

initiation rates $\alpha$ consistent with these patterns of transcriptional upregulation. Further, unlike transcription factor dynamics or other shorter time-scale processes traditionally considered in two state models of transcription regulation, these nucleosome mediated methods of transcriptional control can span timescales from days to weeks giving long term control [73].

Last, in the final section of the manuscript, the proposed ddTASEP was extended to consider both static site defects that introduce variability in the RNAPII nominal advance rate and localized variability in the nucleosome rate constants that also induced pausing. Our results demonstrated that the nucleosome dynamics could be tuned individually to induce localized pausing effects that may occur at genetic points of interest like CpG islands or exons without introducing discontinuous shock profiles that would arise from treating these as static defects. Further, the resulting throttling effects on the flux were comparable if not stronger than those obtained in the static defect case. Additionally, the capacity to modify the nucleosome dynamics along the gene body provides an interesting capability in this model that is not attainable in a two-state transcription model. Utilizing our model, it may be possible to tune the nucleosome dynamics in different regions to serve different functions. For instance, a unique set of wrapping dynamic rate constants could be assigned to the promoter-proximal region to simulate promoter associated histone modifications like H3K9me3 silencing or H3K27ac activation while the gene body could have a separate set of modifications such as H3K79me3 or H4K20me1 promoting transcriptional elongation on the gene body leading to locally elevated polymerase density near the transcription start site with a homogeneous gene body (e.g. see Akhtar et al.) comparable to our density profiles in Fig 9C [19,74,75].

## Supporting information

**S1 Fig. Validity of lattice coarse-graining.** (A) compares the analytical results for the expected waiting time to enter a nucleosome between the single site and fifty site model showing that the results are mathematically identical. (B) compares the analytical results for the variance of the waiting time to enter a nucleosome between the single site and 50 site models confirming that they converge to each other in the limit as $h_c$, $h_o \rightarrow 0$. (A) and (B) are log-log plots with $h_c$ set to {**0.001, 0.01, 0.1, 1**} corresponding to red, blue, yellow, and purple markers respectively with $h_o$ adjusted to achieve the desired values of $E_e$ and $V_e$. (C) Pseudo-color plots of the percent contribution of the bare DNA passage time to the mean and variance of first passage time to clear a nucleosome unit $E_h$ (left) and $V_h$ (right) for the single site (top) and 50 site (bottom) models. (TIF)

## Author Contributions

**Conceptualization:** Robert C. Mines, Tomasz Lipniacki, Xiling Shen.

**Formal analysis:** Robert C. Mines, Tomasz Lipniacki.

**Funding acquisition:** Tomasz Lipniacki, Xiling Shen.

**Software:** Robert C. Mines.

**Supervision:** Tomasz Lipniacki, Xiling Shen.

**Validation:** Tomasz Lipniacki.

**Visualization:** Robert C. Mines.

**Writing – original draft:** Robert C. Mines, Tomasz Lipniacki.

**Writing – review & editing:** Robert C. Mines, Tomasz Lipniacki, Xiling Shen.

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
