## [Decision Letter · Decision Letter 0]

25 Aug 2021

Dear Prof. Shen,

Thank you very much for submitting your manuscript "Slow Nucleosome Dynamics set the Transcriptional Speed Limit and Induce RNA Polymerase II Traffic Jams and Bursts" for consideration at PLOS Computational Biology.

As with all papers reviewed by the journal, your manuscript was reviewed by members of the editorial board and by several independent reviewers. In light of the reviews (below this email), we would like to invite the resubmission of a significantly-revised version that takes into account the reviewers' comments.

We cannot make any decision about publication until we have seen the revised manuscript and your response to the reviewers' comments. Your revised manuscript is also likely to be sent to reviewers for further evaluation.

Sincerely,

Alexandre V. Morozov, Ph.D.

Associate Editor

PLOS Computational Biology

Mark Alber

Deputy Editor

PLOS Computational Biology

Reviewer's Responses to Questions

**Comments to the Authors:**

Reviewer #1: The article explains an interesting phenomenon of appearance of traffic jams on molecular highways as a result of slowing down of nucleosome transcriptional rates, which is an undesirable situation. The model used is ddTASEP which is one of the recent agent based models for studying gene transcription process. Overall the work is nice.

I recommend the article for publication in this journal.

Reviewer #2: Mines et al. study the impact of nucleosome unwrapping on transcriptional dynamics. They introduce a stochastic model of particles (i.e. the RNA polymerase) moving on a lattice (i.e. the gene sequence), where the local speed also stochastically vary between two values (accounting for nucleosome wrapping/unwrapping). This model can be generically formulated as a variant of the TASEP model on an open lattice with dynamic defect (ddTASEP). Mines et al. mainly use simulations to simulate the properties of the model and study the impact of the parameters on the observed dynamics. They notably show how transcriptional bursting can be controlled by the nucleosome dynamics under this model, analytically study the first moments of the first passage time a particle to complete transcription and upon introducing a parameter (gamma) that accounts for the average rate of elongation without particle obstruction, they estimate the role of gamma on explaining the flux and density at equilibrium with a partial phase diagram, using simple heuristics and numerical simulations.

To my knowledge, the theoretical study of the role of nucleosomes in transcriptional pausing and bursting is new and interesting, as an alternative to other models proposed that notably focus on initiation control mechanism. However, I have major concerns about the definition of the model and technical aspects of the paper, which make me doubtful about its relevance in the study of transcription dynamics.

Major comments

1. The authors map the particle occupancy to 1 site, by scaling the transcription bubble to the unit length of the lattice in the TASEP. In reality, RNA polymerases move site by site with a rule of excluding the move if another polymerase is located L nt further, where L is the size of the bubble. These dynamics obey that of a TASEP with particles of extended size that has been studied in the context of translation and transcription (see Zia et al. for a review https://doi.org/10.1007/s10955-011-0183-1 and many other papers, included those cited below). It is also to be noted that at least two recent papers which I am aware of on this topic, e.g. Zuo and Chou arXiv:2103.05151 , and Turowski et al. https://doi.org/10.1016/j.molcel.2020.06.002, use a TASEP model with extended particle size to model transcription. The approximation that consists of scaling a sum of moves with exponential rates to a single exponential jump is a priori wrong (a sum of independent exponential r.v.s is not an exponential r.v.), so the authors need to clarify why this simplification should be still relevant.

2. The aforementioned papers of Zuo and Chou and Turowski et al. especially focus on some important aspects of RNA Pol dynamics, including backtracking and DNA torque. While these studies don’t require or assume a nucleosome free DNA in the modelling, I believe that the present paper is strongly limited in the biological interpretation, as there is no backtracking or heterogeneity in transcription (for example GC content biases the elongation rate), which are known to affect transcription dynamics. With the orders of magnitude involved, is it reasonable to consider a TASEP model that does not take these aspects into account?

3. While the authors investigate the size and frequency of transcriptional bursts, I find surprising that they don’t relate their study to the widely used model of transcriptional burst induced by two state promoter (see review by Sanchez and Golding DOI: 10.1126/science.1242975 , or this more recent review doi: 10.1111/brv.12452 ), with many experimental papers working with this framework (including recent papers such as DOI: 10.1126/sciadv.aaz6699 or doi : 10.1038/s41467-021-24461-6) to explain burst size and frequency. While this mechanism controls the burst at initiation, does an elongation-controlled burst yields any difference, or could be better explaining some experimental observations?

4. In the section where the parameter gamma is introduced and studied, as the modified elongation rate due to histone dynamics, I find the derivation of the mean and variance for the FPT not very interesting (it is a quite basic calculation using continuous time Markov chain); the relevance of this part is stated at the very end of the paper, but with no evidence or reference that would support the authors claim, so I think this should either be a supplement, or revised to make it more relevant with actual experiments and applications.

5. Technically, the authors provide some heuristics to infer the flux and densities that yield a partial phase diagram in gamma. While the results are sound, I also suspect they can also be quite naturally derived from a mean-field approximation to solve the master equation associated with the process (some formulas seem quite close to the TASEP with extended particles, with some correspondence between the parameters gamma and the particle size DOI:10.1088/0305-4470/36/8/302). This would confirm the authors result and help to interpret the limitations that seem to numerically appear in a certain range of the parameters.

Minor comments

1. There should be a rigorous mathematical definition of the ddTASEP model in the methods sections

2. Fig 4: In the IL region, typo on J

**Have the authors made all data and (if applicable) computational code underlying the findings in their manuscript fully available?**

Reviewer #1: Yes

Reviewer #2: Yes

PLOS authors have the option to publish the peer review history of their article (what does this mean?). If published, this will include your full peer review and any attached files.

Reviewer #1: **Yes: **Isha Dhiman

Reviewer #2: No
---

## [Decision Letter · Decision Letter 1]

6 Jan 2022

Dear Prof. Shen,

We are pleased to inform you that your manuscript 'Slow Nucleosome Dynamics set the Transcriptional Speed Limit and Induce RNA Polymerase II Traffic Jams and Bursts' has been provisionally accepted for publication in PLOS Computational Biology.

Best regards,

Alexandre V. Morozov, Ph.D.

Associate Editor

PLOS Computational Biology

Mark Alber

Deputy Editor

PLOS Computational Biology

Reviewer's Responses to Questions

**Comments to the Authors:**

Reviewer #1: I recommend this paper for publication in this journal.

Reviewer #2: The authors thoroughly answered the points raised in my review and strengthened their manuscript while opening some very interesting lines for future work. I am happy to recommend it for publication in PLoS Comp Bio

**Have the authors made all data and (if applicable) computational code underlying the findings in their manuscript fully available?**

Reviewer #1: Yes

Reviewer #2: Yes

PLOS authors have the option to publish the peer review history of their article (what does this mean?). If published, this will include your full peer review and any attached files.

Reviewer #1: **Yes: **ISHA DHIMAN

Reviewer #2: No

---

## [Editor Report · Acceptance letter]

2 Feb 2022

PCOMPBIOL-D-21-01235R1 

Slow Nucleosome Dynamics set the Transcriptional Speed Limit and Induce RNA Polymerase II Traffic Jams and Bursts

Dear Dr Shen,

I am pleased to inform you that your manuscript has been formally accepted for publication in PLOS Computational Biology. Your manuscript is now with our production department and you will be notified of the publication date in due course.

With kind regards,

Anita Estes
